# Mapping lesion-specific response and progression dynamics and inter-organ variability in metastatic colorectal cancer

Jiawei Zhou[1], Amber Cipriani [1,2], Yutong Liu[3], Gang Fang[4], Quefeng Li[3] & Yanguang Cao [1,5] ✉

Achieving systemic tumor control across metastases is vital for long-term patient survival but remains intractable in many patients. High lesion-level response heterogeneity persists, conferring many dissociated responses across metastatic lesions. Most studies of metastatic disease focus on tumor molecular and cellular features, which are crucial to elucidating the mechanisms underlying lesion-level variability. However, our understanding of lesion-specific heterogeneity on the macroscopic level, such as lesion dynamics in growth, response, and progression during treatment, remains rudimentary. This study investigates lesion-specific response heterogeneity through analyzing 116,542 observations of 40,612 lesions in 4,308 metastatic colorectal cancer (mCRC) patients. Despite significant differences in their response and progression dynamics, metastatic lesions converge on four phenotypes that vary with anatomical site. Importantly, we find that organ-level progression sequence is closely associated with patient long-term survival, and that patients with the first lesion progression in the liver often have worse survival. In conclusion, our study provides insights into lesion-specific response and progression heterogeneity in mCRC and creates impetus for metastasis-specific therapeutics.

Metastasis is the leading cause of cancer mortality[1]. Unfortunately, antitumor therapies are still designed mostly based on the biology of primary tumors, with little consideration of metastases[2,3]. Achieving systemic tumor control across metastases is critical for long-term survival but remains intractable in many patients. Some metastases respond highly to treatment while others do not at all, resulting in many dissociated and heterogeneous responses within patients[4–7]. Lesion-level response and progression heterogeneity are common in many cancer types, but our understanding of such lesion-level response heterogeneity and its relevance to prognosis remains rudimentary.

Most investigations of metastatic heterogeneity focus on tumor genetic mutations, clonal compositions, or transcriptomics[8–10]. These molecular and cellular characterizations are critical to elucidating the underlying mechanisms of lesion response heterogeneity[11,12]. However, it is equivalently critical to study metastatic heterogeneity on the macroscopic level, such as distinct lesion dynamics in growth, response, and recurrence during treatment, as well as their potential phenotypic convergence anatomically. These phenotypes would complement molecular and cellular analyses for a holistic view of lesion-specific heterogeneity. The local microenvironment selects tumor phenotypes in response to treatment, leading to heterogeneity

[1]Division of Pharmacotherapy and Experimental Therapeutics, School of Pharmacy, University of North Carolina at Chapel Hill, Chapel Hill, NC 27599, USA. [2]UNC Health Medical Center, Department of Pharmacy, Chapel Hill, NC 27514, USA. [3]School of Public Health, University of North Carolina at Chapel Hill, Chapel Hill, NC 27599, USA. [4]Division of Pharmaceutical Outcomes and Policy, School of Pharmacy, University of North Carolina at Chapel Hill, Chapel Hill, NC 27599, USA. [5]Lineberger Comprehensive Cancer Center, School of Medicine, University of North Carolina at Chapel Hill, Chapel Hill, NC 27599, USA. ✉e-mail: yanguang@unc.edu

across anatomically distinct lesions in terms of response and progression dynamics[13,14]. Characterizing their phenotypic differences (divergence) or similarities (convergence) could yield insights into tumor ecological features and systemic resistance. The spatiotemporal patterns of response and progression at the lesion or organ level could not only be informative to prognosis, but also could enrich our knowledge of metastasis-to-metastasis interactions and the systemic consequence of regional progression. This study sought to investigate spatiotemporal response heterogeneity through mapping lesion-specific response and progression dynamics in metastatic CRC (mCRC).

Colorectal cancer (CRC) is the third leading cause of cancer-related death[15]. About 20% of CRC patients have distant metastases at diagnosis; the five-year relative survival rate is only 14% for these patients[16,17]. Lesion-specific response heterogeneity is common in CRC patients treated with either standard chemotherapy alone or in combination with targeted therapy[18]. We, along with others, have found that high response heterogeneity is associated with worse survival[18–21]. Importantly, we also found favorable responses in liver metastases predicted longer patient survival, compared to lesions in the lungs and lymph nodes (LN)[18]. Characterizing lesion-specific response heterogeneity in mCRC is therefore valuable for prognosis and therapies.

In this study, to map the lesion-level response and progression patterns in mCRC, we first apply a mathematical model to capture tumor growth dynamics in 4,308 mCRC patients. Next, individual lesion-specific response and progression probabilities are mapped to predict their phenotypic divergence and convergence across anatomical sites. Last, we apply a machine learning approach to analyze the progression sequence across organs and its relevance to long-term patient survival. The spatiotemporal patterns of response and progression at the organ level could infer tumor evolution in space and time, affording more biological hypotheses. Our study provides insights into lesion-specific phenotypic heterogeneity in mCRC and yields substantial implications for designing metastasis-specific therapeutics.

## Results

### Data sources and structure

To evaluate metastatic response and progression dynamics in mCRC, we collected longitudinal radiographical measurements of metastatic lesions in colorectal cancer (CRC) patients from Project Data Sphere. In total, 4308 patients with 40,612 lesions from eight Phase III trials were included. The inclusion and exclusion criteria are presented in Fig. 1a. The distribution of lesion number across organs is shown in Fig. 1b. The total target lesions were 19,180 with 94,174 radiographic measurements by CT scan, and there were 18,594 nontarget lesions and 2838 new lesions that had records of response status, appearance time, and anatomical site, and these lesions did not have longitudinal radiological measurements. Additional information including patients' demographic and clinical characteristics (e.g., age, gender, race, body mass index [BMI], tumor type, treatment history, RECIST response, and *KRAS* status), progression-free survival (PFS) and overall survival (OS) are reported in Table 1. We also included the tumor longitudinal measurements in a head and neck squamous cell carcinomas (mHNSCC) trial for an external validation. The data was also from Project Data Sphere with similar inclusion/exclusion criteria as in the CRC data.

### Model recapitulated tumor growth dynamics of individual lesions

The tumor growth dynamics of 19,180 target lesions with 94,174 radiographical measurements were fitted using a tumor growth model[22]. There are three dynamic parameters in the model: the regression (tumor-killing) rate $Kd$, the fraction of non-responding (or resistant) cells $F$, and the progression (or regrowth) rate $Kg$ (Fig. 2a). The model was optimized using a nonlinear mixed effect (NLME) modeling approach, which allows the estimation of three dynamic parameters at the individual level and their inter-lesion variance within the population. Overall, the model adequately recapitulated the longitudinal profiles of tumor radiographic measurements for each lesion. The goodness-of-fit and model visual predictive check plots, as

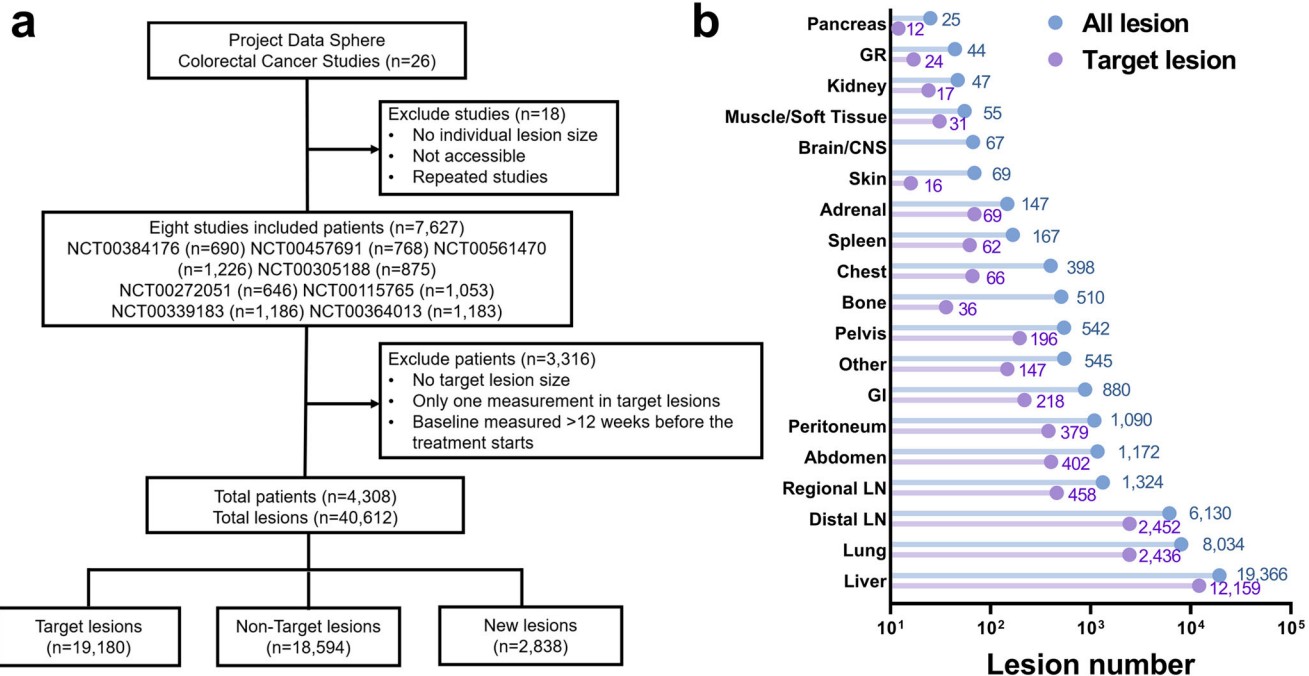

**Fig. 1 | Data source. a** CONSORT diagram of metastatic colorectal cancer data inclusion and exclusion criteria. **b** The number of all lesions (target, non-target, and new) and target lesions across organs. GR Genitourinary and Reproductive, CNS Central nervous system, GI Gastrointestinal tract, LN Lymph nodes. Source data are provided as a Source Data file.

**Table 1 | Demographic information of colorectal cancer patients**

| Variable | | Variable | |
|---|---|---|---|
| Age, years (mean, sd) | 60.2 (10.8) | Metastatic organ number (n, %) | |
| Gender (n, %) | | 1 | 553 (12.8) |
| Male | 2538 (58.9) | 2 | 1159 (26.9) |
| Female | 1770 (41.1) | 3 | 1146 (26.6) |
| Self-Reported Race (n, %) | | ≥ 4 | 1450 (33.7) |
| White/Caucasian | 3883 (90.1) | Metastatic lesion number per patient (median, IQR) | 9 (6, 12) |
| Black/African American | 104 (2.4) | Number of radiographic measurements per target lesion (median, IQR) | 4 (3, 6) |
| Asian | 142 (3.3) | KRAS status (n, %) | |
| Other | 179 (4.2) | Wild-Type | 795 (18.4) |
| Body Mass Index, kg/m$^2$ (mean, sd) | 26.2 (5.1) | Mutant | 593 (13.8) |
| Tumor Type (n, %) | | Unknown | 2920 (67.8) |
| Colon | 2581 (59.9) | Response (n, %) | |
| Rectal | 1359 (31.5) | Complete Response | 118 (2.7) |
| Unspecified | 368 (8.5) | Partial Response | 1473 (34.2) |
| Prior Surgery (n, %) | | Progressive Disease | 781 (18.1) |
| Yes | 2993 (69.5) | Stable Disease | 1806 (41.9) |
| Primary tumor resected | 2663 (61.2) | Not Evaluable | 130 (3) |
| Liver metastases resected | 261 (6.05)[a] | Treatment[b] (n, %) | |
| Lung metastases resected | 27 (0.6)[a] | Bevacizumab plus chemotherapy | 376 (8.7) |
| LN metastases resected | 206 (4.8)[a] | Bevacizumab plus FOLFOX | 630 (14.6) |
| No | 1315 (30.5) | FOLFIRI alone | 1303 (30.2) |
| Prior Radiation (n, %)[c] | | FOLFOX alone | 762 (17.7) |
| Yes | 445 (10.3) | Panitumumab plus Bevacizumab plus chemotherapy | 372 (8.6) |
| No | 3345 (77.6) | Panitumumab plus FOLFOX | 441 (10.2) |
| Unknown | 518 (12.1) | Panitumumab plus FOLFIRI | 424 (9.8) |
| Progression free survival, days (median, IQR) | 247 (241.5) | Overall survival, days (median, IQR) | 443 (410.3) |

[a]Study NCT00272051 and NCT00305188 did not provide the location of the resected metastases and were not included in the liver/lung/lymph nodes metastases resected patient number calculations.

[b]FOLFOX is the combination of folinic acid, fluorouracil and oxaliplatin. FOLFIRI is the combination of folinic acid, fluorouracil and irinotecan. *Sd* standard deviation, *LN* lymph nodes, *IQR* Interquartile range.

[c]Radiation was applied to rectal cancer and some colon cancer with lung metastases.

well as representative individual fittings, show good model predictive performance (Supplementary Fig. 1).

Population estimates and inter-lesion variances in tumor dynamic parameters are summarized in Supplementary Table 1. The parameters for individual lesions significantly differed across organs ($p < 0.0001$, Fig. 2b). Among all metastases, lesions in the bone exhibited the lowest tumor shrinkage (1-*F*), while lesions in the genitourinary and reproductive (GR) system had the fastest progression rates (*Kg*), and kidney lesions showed the lowest regression rates (*Kd*). Among three most abundant metastatic sites (liver, lung, and LN), lesions in the liver showed the highest tumor shrinkage but the fastest progression rates, suggesting the unique response feature of liver lesions.

Higher fractions of treatment-resistant cell (*F*) is associated with slower rates of regression (*Kd*, $r = -0.69$, $p = 0.0014$) and faster rates of progression (*Kg*, $r = 0.53$, $p = 0.03$, Fig. 2c). Progression rates seemed to be independent of regression rates (Fig. 2c). Remarkably, no significant correlations were observed between baseline tumor burden and all tumor dynamic parameters (Fig. 2d). Large tumor burden, on the individual lesion level, did not necessarily confer slow regression rates, high treatment-resistant fractions, or slow progression rates, implying that tumor burden at baseline is not a robust prognostic factor in mCRC[23–25]. Notably, metastatic lesions under antibody targeted therapy (bevacizumab and/or panitumumab) plus chemotherapy (FOLFOX or FOLFIRI), compared to standard chemotherapy alone, showed significantly deeper response (effect size = 0.43) and lower progression rates (effect size = 0.26), but had a moderate effect on tumor regression rates (effect size = 0.06, Supplementary Fig. 2).

**Response and progression dynamics suggest phenotypic convergence on the organ level**

The tumor growth model predicted the longitudinal profiles of response and progression for each target lesion. Time to response and progression were then derived as the duration from the start of treatment to the time of response or progression per RECIST v1.1[26], respectively. We compared our model-predicted lesion response and progression rates with patient response status per RECIST 1.1. Complete (CR) or partial responders (PR) had shorter time to response and more extended duration before progression than patients with stable disease (SD). (Supplementary Fig. 3).

We integrated the time to response for both target and non-target lesions and the time to progression for all lesions, including the new ones, into random effect Cox proportional models[27]. The Cox model predicted the relative probabilities of lesion response or progression at the organ level. We tested the covariate effects of treatment, age, BMI, gender, race, surgical history, and line of therapy (first or second line) in the random effect Cox proportional model. The covariate effects were summarized in Supplementary Fig. 4a and Supplementary Fig. 4b. Significant covariates were included in the final model. Of note, treatment, surgical history, and line of therapy were included as significant covariates in the progression model, and treatment, race, and line of therapy were included in the response model.

With these covariate effects, we could more accurately estimate organ-intrinsic response and progression characteristics. The hazard ratios for the response and progression across organs are shown in Fig. 3a and Fig. 3b. With abdominal lesions as the reference, metastatic

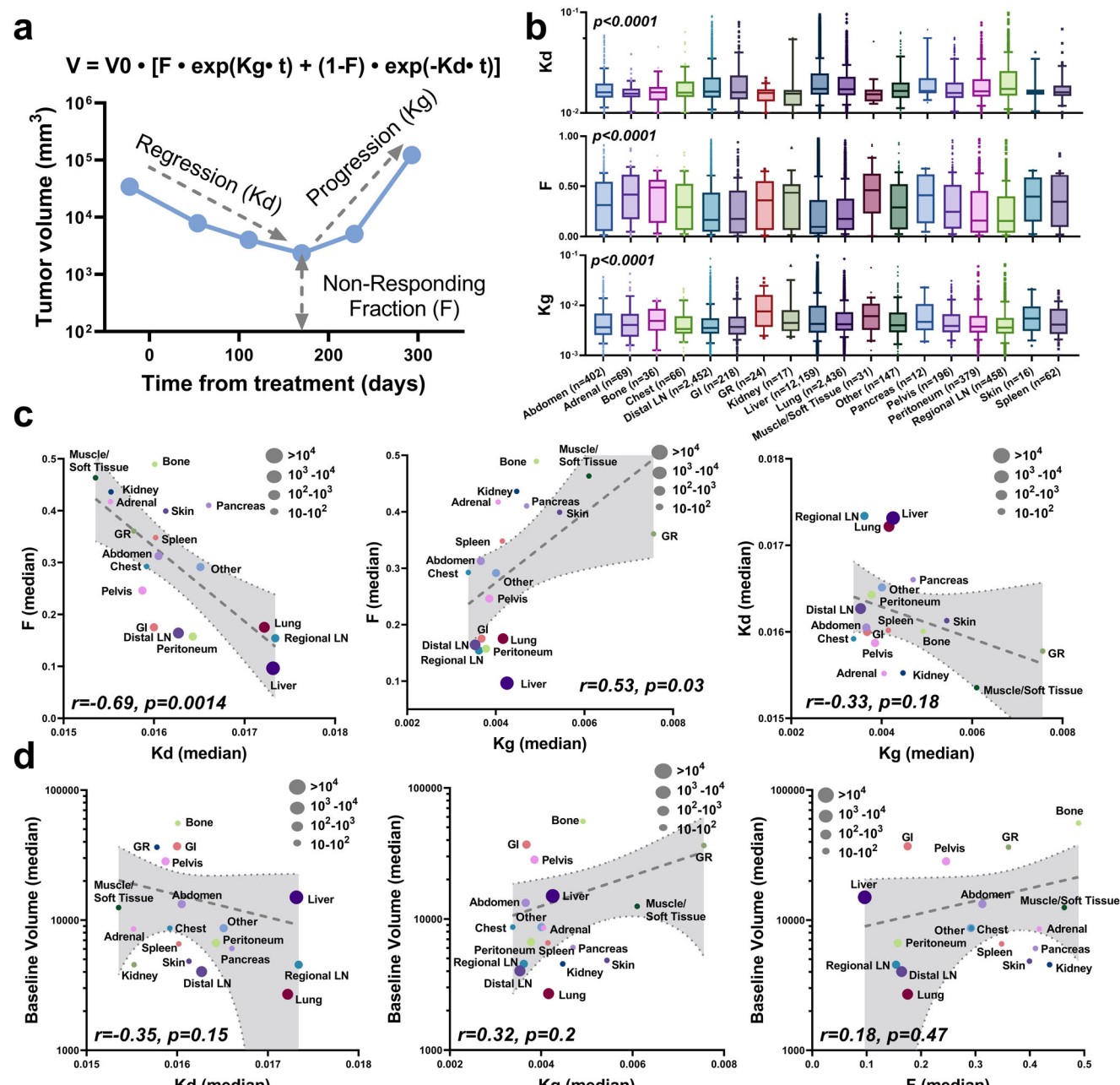

**Fig. 2 | Tumor response dynamics were recapitulated by modeling. a** Schematic plot of tumor growth model. **b** Box plots of model parameters *Kd, F* and *Kg* across organs. Significance was calculated using Kruskal-Wallis tests. The box extends from the 25th to 75th percentiles and the line in the middle is plotted as the median. The whiskers are drawn down to the 10th percentile and up to the 90th percentile. Points below and above the whiskers represent individual lesions. **c** The correlations between model parameters. **d** The correlations between model parameters and tumor baseline volume. The size of the dots represents lesion number (reported in **b**). The dashed lines with gray area are the linear regression with 95% confidence interval. The correlation coefficients and *p*-values were calculated using two-tailed Pearson correlation tests. Source data are provided as a Source Data file.

lesions in the liver were most likely to respond to treatments, whereas lesions in the brain/central nervous system (CNS) were least likely (Fig. 3a). Lesions in the gastrointestinal (GI) system, skin, and bone were significantly less likely to respond than abdominal lesions. Lesions in the spleen, lung, and peritoneum showed comparable responses. The probability of progression also differed greatly across anatomical sites (Fig. 3b). The metastatic lesions with the highest likelihood of progression were those in the brain/CNS, GR system, and liver, while lesions in the GI system, and regional and distal LNs were least likely.

We then integrated organ-specific response and progression probabilities to investigate their potential phenotypic convergence across anatomical sites. As in Fig. 3c, an anatomical chart of organ-specific response and progression probabilities was created based on their relative hazards in the Cox model. Four types of phenotypic features emerge in CRC-metastatic organs defined by their associated lesions' likelihood of response and progression. Notably, bone and brain lesions had low response and high progression probabilities (low-high phenotype), while liver lesions had high probabilities of both response and progression (high-high phenotype). Patients with these metastases, particularly those with low-high phenotype, had much worse survival outcomes than those with other phenotypes (OS median 378 days vs. 561 days, *p* < 0.0001, Supplementary Fig. 5a). On the other side, metastatic lesions in the lung and LN showed high response

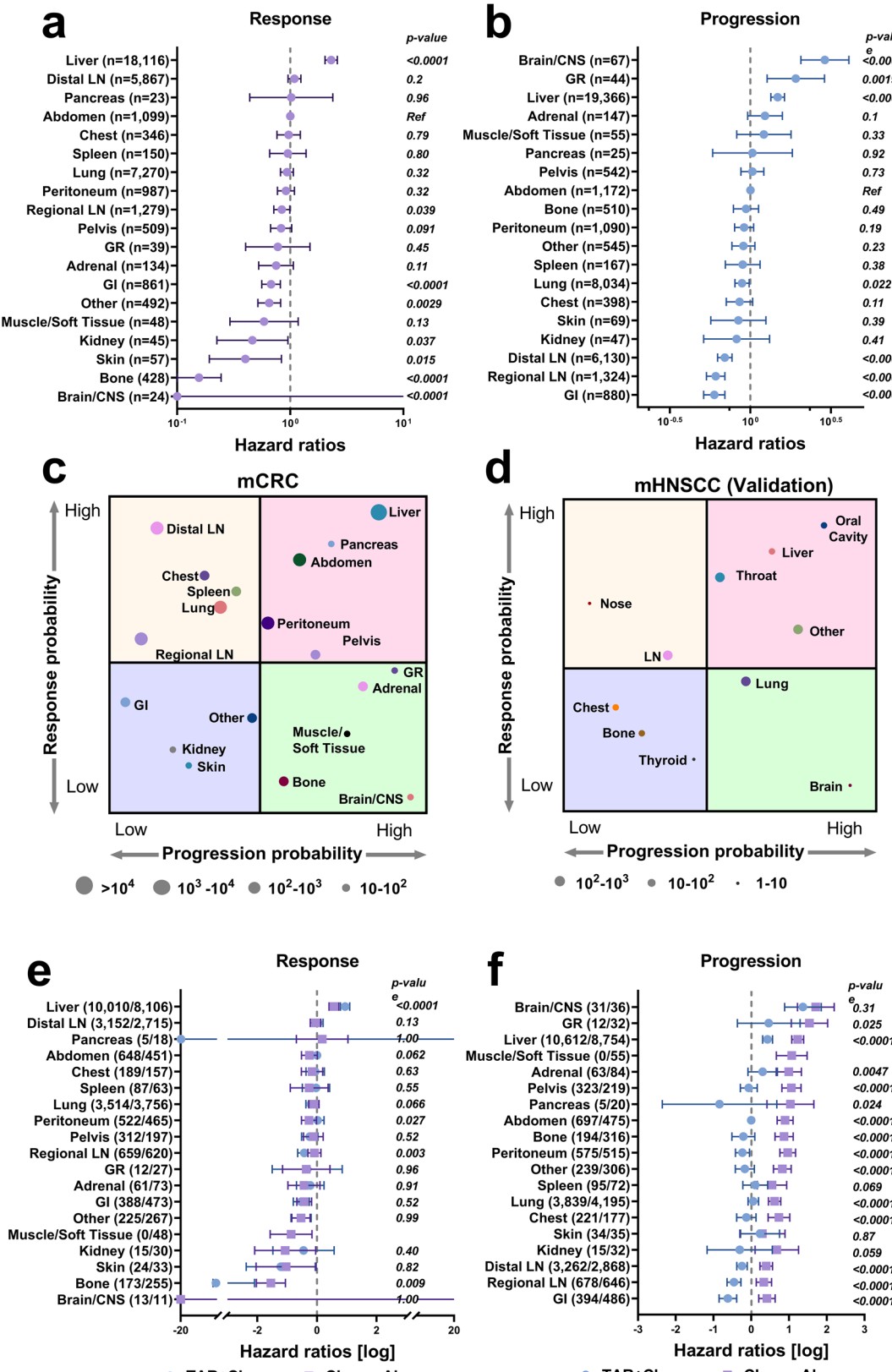

**Fig. 3 | Organ-level tumor response and progression probabilities suggest phenotypic convergence. a**, **b** Data are presented as the hazard ratio estimates with 95% confidence interval by organs on lesion response and progression in colorectal cancer patients. **c**, **d** are the anatomical charts of organ-specific response and progression hazard ratios in metastatic colorectal cancer (mCRC) and metastatic head and neck squamous cell carcinomas (mHNSCC). **e**, **f** Data are presented

as the hazard ratio estimates with 95% confidence interval in response and progression by organs stratified on treatments in mCRC. *P*-values in **a**, **b**, **e**, and **f** were calculated by two-sided likelihood ratio tests. TAR + Chemo, antibody targeted therapies (bevacizumab or panitumumab) plus chemotherapy; Chemo Alone, chemotherapy alone. Source data are provided as a Source Data file.

and low progression probabilities (high-low phenotypes). Patients who have metastases in high-low phenotype organs only tend to have a better prognosis than patients with other phenotypic metastases do (OS median 770 days vs. 524 days, $p < 0.0001$, Supplementary Fig. 5b).

Interestingly, most metastatic lesions with high progression probabilities tend to occur in organs known to have immunosuppressive microenvironments, such as the liver, bone, and brain/CNS[28–31]. To show the anatomical pattern of lesion response and progression is beyond tumor biology and is more closely related to tissue microenvironments, we performed a validation analysis in a biologically distinct cancer type, head and neck squamous cell carcinomas (mHNSCC), to see whether a similar anatomical chart exists (Fig. 3d). A total of 393 patients with 1892 lesions were analyzed, including eleven metastatic organs (Supplementary Fig. 6a, b). Patients' demographics are reported in Supplementary Table 2. We built random effect Cox proportional models to estimate hazard ratios across organs in mHNSCC, as we did in mCRC. Treatment, age, radiation history, and body surface area (BSA) were included as covariates in the progression model. Treatment and race were considered as covariates for the response model. The organ-specific hazard ratios for lesion progression and response were shown in Supplementary Fig. 6c, d. In mHNSCC, metastases in the liver, and brain also showed high progression potential, in line with what we observed in mCRC. Metastatic lesions in the LNs exhibit a high-low phenotype, consistent with mCRC. Similar anatomical charts across cancer types suggest that organ-intrinsic microenvironmental factors, such as the local physical and immunological components, could be key modulators to the mechanisms underlying the probabilities of tumor response and progression. However, further investigations are warranted.

In mCRC, treatment effects on organ-specific responses were also investigated. For simplicity, treatments were divided into two groups, chemotherapy alone and in combination with antibody targeted therapy. The combined antibody targeted therapies are either panitumumab or bevacizumab, or both. Surprisingly, combination with the antibody targeted therapies did not significantly influence organ-specific response probabilities (Fig. 3e), suggesting limited direct cytotoxic effects of antibody-based targeted therapies. Notably, the primary therapeutic benefit of antibody targeted therapies was to decrease lesion progression (Fig. 3f). Progression hazards significantly decreased in most metastatic organs except for the skin, brain/CNS, spleen, and kidney. Taken together, antibody-targeted therapies showed effect primarily on decreasing lesion progression and had limited influence on the lesion response probability. Interestingly, cytotoxic chemotherapies did not seem to influence lesion progression patterns, and high-progression organs in Fig. 3c also had remained to have high progression probability during cytotoxic chemotherapies (Fig. 3f), reiterating a critical role for local tissue environments in long-term tumor control.

## Progression sequence across organs predicts patient survival

We built a k-means unsupervised clustering model[32] to cluster patients based on their organ-level lesion progression sequence and investigate their relevance to patient survival. Elbow sum of square[33] (Supplementary Fig. 7a) and Silhouette score[34] (Supplementary Fig. 7b) were calculated to determine the optimal k in the final classification. Akaike information criterion (AIC) and Bayesian information criterion (BIC)[35] were also applied to find optimal k (Supplementary Fig. 7e). $K = 4$, 5, and 6 showed similar performance in the model evaluation metrics. The patient survival profiles were also compared using concordance, and the select $k = 5$ showed the finest separation of patient survival, resulting in distinct features of progression for each group. Five groups of patients were thus identified with distinct patterns of organ-specific progression sequences and were stratified by progressive organ number and first-progressive organ: Mono-Organ ($n = 1425$), Hetero-Organ ($n = 801$), Lung-First ($n = 577$), Liver-First ($n = 1194$), and

the Other-First ($n = 888$) groups. The clinical demographics and baseline information of each group are summarized in Supplementary Table 3.

Organ-level progression sequence is significantly correlated with long-term patient survival ($p < 0.0001$, Fig. 4b). As expected, patients with multiple organ progression had worse survival than patients with only one organ progression (OS median Hetero-Organ 385 days vs. Mono-Organ 653 days). Remarkably, despite comparable number of baseline metastases, patients whose first progression was in the liver had a much worse prognosis than those whose first progression was in lungs or other sites (OS median Liver-First 450 days vs. Lung-First 679 days vs. Other-First 581 days, Fig. 4b and Supplementary Fig. 8). This is consistent with earlier observations (Fig. 3c) that lesions in the lung had high-low phenotype that is often associated with good patient prognosis. Patients with tumor progression first in the liver had faster subsequent progression than patients whose progression occurred in lungs or other sites, suggesting that progressive lesions in the liver may have high systemic consequences ($p < 0.0001$, Fig. 4c). It also aligns with our previous finding that the response of liver lesions to treatments strongly predicted patient survival[18].

Next, we performed k-means unsupervised clustering in the Hetero-Organ group to further investigate progression patterns in patients with extensive metastases progression. Four groups of patients were optimally clustered (Supplementary Fig. 7c, d, f), and one distinctive feature among these clusters was the progression order of liver lesions (Supplementary Fig. 9a). Despite similar baseline metastases, patients with first or second progression occurring in the liver had worse survival than those with early progression occurring in other organs (Supplementary Fig. 9b), but two groups showed no significant difference in subsequent time to progression (Supplementary Fig. 9c). This observation further underlines the importance and systemic consequence of liver lesions to tumor response and resistance.

## Targeted antibody therapies minimally influence lesion progression sequence

We compared the progression sequence in patients under different treatments (chemotherapy alone vs. combination with antibody targeted therapy). In patients with Liver-First, Lung-First or Other-First progression patterns, antibody targeted therapies significantly improved patient overall survival ($p < 0.0001$, Fig. 5a). However, neither the proportion of patients with each progression cluster (Fig. 5b) nor the sequence of progression across metastatic organs were significantly changed by antibody targeted therapies (Fig. 5c–e). Tumor progression in the GR and pancreas occurred slightly earlier in antibody targeted therapy, which did not seem to translate meaningful difference in patient survival. Despite similar sequences, patients under antibody targeted therapies had significantly slower first and second progression, even without significant difference in the third or later progression (Fig. 5f, g). The average time to progression were much longer in combination therapy compared to chemotherapy alone.

In patients with the Hetero-Organ pattern, antibody targeted therapies did not meaningfully improve overall survival (Supplementary Fig. 10a) compared to chemotherapy alone, and the proportions of patients in each subcluster were similar between the two treatment groups (Supplementary Fig. 10b). Patients' progression patterns and lesion time to progression were largely comparable, especially for those who had early liver lesion progression (Supplementary Fig. 10c–h). Similarly, antibody targeted therapies did not influence lesion progression sequence. Overall, the primary therapeutic benefit of antibody targeted therapies was to delay progression in patients with few ($< 4$) metastatic organs, but not in those with broad metastases.

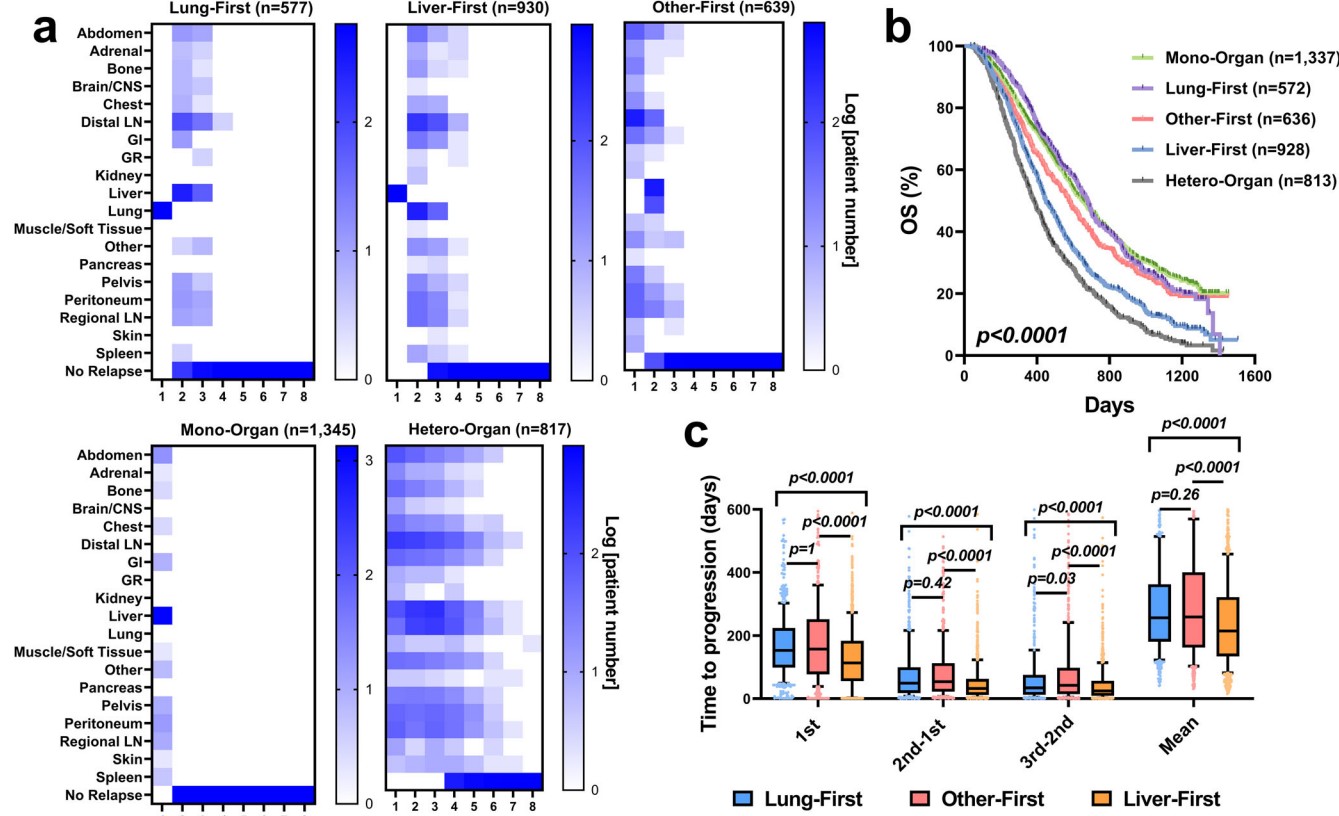

**Fig. 4 | Patient progression sequence association with patient survival.**
**a** Patients were clustered into five groups based on their lesion progression sequence. The column labels are the progression sequence. Color of the heatmap represents the log10 scale of patient number (all plus one to avoid zero values). **b** Kaplan-Meier curves of clustered patients overall survival. **c** Box plots of the first lesion progression time (1st), time between first and second progression (2nd-1st), time between second and third progression (3rd-2nd), time between third and fourth progression (4th-3rd), and the average progression time in Lung-First (*n* = 577), Other-First (*n* = 639), and Liver-First (*n* = 930). The box extends from the 25th to 75th percentiles and the line in the middle is plotted as the median. The whiskers are drawn down to the 10th percentile and up to the 90th percentile. Points below and above the whiskers represent individual lesions. *P*-values in **c** were calculated by two-sided Dunn's multiple comparisons. Source data are provided as a Source Data file.

## Machine learning model predicts lesion progression sequence

In order to predict patient progression sequence at the time of diagnosis, we built a gradient boosting model using patient baseline characteristics and metastases profiles[36]. The model parameters are in Supplementary Table 4. The area under the receiver operating characteristic (ROC) curve of the testing data was 0.91, which indicated fair performance (Supplementary Fig. 11a). The model could predict Mono-Organ and Hetero-Organ groups better than Lung-First, Liver-First, and Other-First groups with higher area under the ROC curve. This indicates that more follow-up information about tumor early response is imperative to predict the progression sequences of the latter three groups (Supplementary Fig. 11b).

## Discussion

Metastasis is responsible for the majority of cancer-related mortality. Unfortunately, systemic tumor control across metastases remains intractable in many patients. This study investigated inter-lesion heterogeneity by analyzing response dynamics of 40,612 lesions to multiple types of treatment in 4308 mCRC patients. Without molecular characterizations of metastases, we focused on the phenotypic features associated with lesion response and progression dynamics as well as the anatomical distributions of these features. Our analyses yielded several intriguing findings. First, metastases differed considerably in their response to treatment, with the tumor shrinkage fraction positively correlating with regression rate and negatively correlating with progression rate. Second, metastatic lesions within the same organ exhibited congruent response and progression dynamics, converging upon four organ-level phenotypes. Metastatic lesions in the liver exhibited high response and high progression probabilities (high-high phenotype), while lesions in the bone and brain/CNS had low response and high progression probabilities (low-high phenotype). These phenotypes appear to be determined by tumor local micro-environments and go beyond tumor biology, as we found a similar pattern in a biologically distinct tumor type, mHNSCC. Third, we found that organ-level progression sequence was closely associated with patient survival, and patients with the first progression in the liver had worse survival outcomes compared to patients with first progression in other organs.

This study quantified the degree of inter-lesion heterogeneity by modeling tumor regression and progression dynamics. By assuming first-order regression of drug-sensitive cancer cells (log-kill hypothesis), the empirical model adequately recapitulated the longitudinal size measurements on the lesion level. The first-order regression implies that drug-sensitive cancer cells may have only one rate-limiting step on the path to cell death[37]. Large tumors are often expected to have tumor regression potentially deviating from strict first-order kinetics because of their non-uniform drug distributions inside the tumor or only the surface tumor cells being actively proliferating and sensitive to treatments[38–40]. Our analyses did not find evidence to support these speculations. Baseline tumor burden did not correlate with tumor regression rates, restating the first-order regression. In contrast, tissue microenvironment matters more than the lesion size to tumor response to treatments. Despite large sizes, metastatic

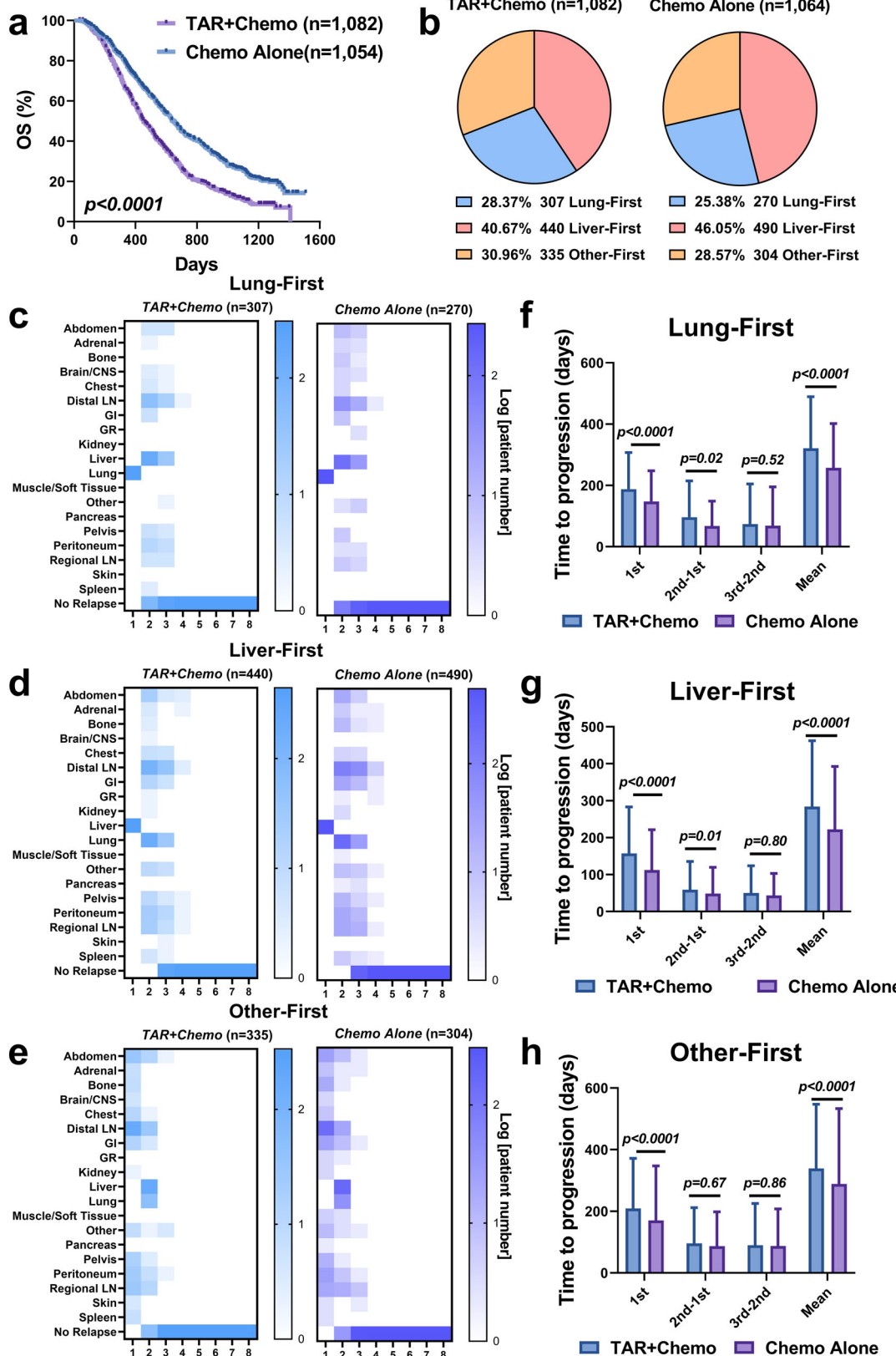

lesions in the liver had relatively higher regression rates than lesions at other organ sites.

Tumor progression rates showed much higher variation during treatment than their associated regression rates and accounted the majority of intrapatient heterogeneity. Lesion progression time, i.e., lead time to progression, was more closely determined by the progression rates rather than the response rate. This finding was in line with Stein et al., who reported that tumor progression rate was a stronger predictor of patient survival[41]. If validated prospectively, the progression rates would offer more appropriate efficacy endpoints in clinical trials than the current ones that focus on the early tumor response and shrinkage, such as overall response rate and best of response.

**Fig. 5 | Targeted therapy decreases average time to progression but has minimal effect on progression sequence. a** Lung-First, Other-First and Liver-First patients overall survival stratified by treatments. **b** Lung-First, Other-First and Liver-First patient proportions by treatments. **c–e** are patient progression sequences stratified by treatments. **f–h** are the box plots of the first lesion progression time (1st), time between first and second progression (2nd-1st), time between second and third progression (3rd-2nd), time between third and fourth progression (4th-3rd), and the average progression time by treatments of the groups in **c–e**. $N = 307/$ $n = 440/n = 335$ patients from TAR + Chemo and $n = 270/n = 490/n = 304$ patients from Chemo Alone were included in **f–h**. The box extends from the 25th to 75th percentiles and the line in the middle is plotted as the median. The whiskers are drawn down to the 10th percentile and up to the 90th percentile. Points below and above the whiskers represent individual lesions. $P$-values in **f–h** were calculated by two-sided Kruskal-Wallis tests. TAR + Chemo, antibody-targeted therapies (bevacizumab or panitumumab) plus chemotherapy; Chemo Alone Chemotherapy alone. Source data are provided as a Source Data file.

Antibody therapies significantly increased tumor shrinkage fraction and showed lesion progression but did not considerably affect tumor-shrinking rates. These observations indicate that the therapeutic benefit of combined antibody therapies is primarily from tumor regrowth suppression rather than direct tumor killing effect. In renal cell carcinomas, bevacizumab significantly reduced tumor regrowth rate, which could become more apparent after progression, in line with our observations in mCRC[42].

Metastatic lesions with a lower fraction of non-responding tumor cells ($F$) also had a slower lesion progression rate. The small fraction of non-responding (resistant) tumor cells prior to treatment implies their low fitness compared to responding (sensitive) populations, which led to low tumor regrowth rates after progression. Interestingly, metastatic lesions in the liver appear to behave differently; they had higher probability to respond, but also faster lesion progression rates than lesions in the LN and lungs, suggesting unique ecological properties of liver lesions. Our analyses highlight the importance of tissue microenvironments to metastatic phenotypes. Metastatic lesions with higher responses were typically found in the liver, spleen, LN, and lungs. These organs are known to have discontinuous or fenestrated endothelial membranes, which may lead to higher drug exposure, potentially conferring high treatment responses[43,44]. In contrast, the organs bearing poorly-responding lesions are usually those with continuous endothelial membranes and thus more limited drug distribution, such as the muscle and brain/CNS[45–48]. Some organs that bear poorly responding metastatic lesions, such as kidney and muscle, have relatively dense tissue matrices. Dense tissue matrix could restrict the growth rate of metastatic lesions[49,50] and also render them less responsive to cytotoxic chemotherapy probably due to limited drug distribution[51,52].

On the other hand, organ-specific progression probabilities seem to closely relate to organ microenvironments. Metastatic lesions with higher progression potentials were often found in the liver, bone, and brain/CNS, which either are immune-privileged or tolerogenic organs[13,14,28–31]. Interestingly, high lesion progression in these organs also occurred during cytotoxic chemotherapies even though such therapies are usually not considered to involve the immune function for therapeutic effect (Fig. 3f). Higher containment effect of tumor progression in immunocompetent organs implies the critical role anticancer immunity plays in long-term tumor control. Patients with highly progressive lesions, such as lesions in the liver and bones, had much worse survival outcomes and these patients likely required more effective and targeted therapeutics.

Tumor progression is a serious impediment to cancer treatment, but organ-level progression patterns remain poorly characterized. We found that early tumor progression in the liver, compared to early progression in other sites, predicts worse patient survival and more rapid subsequent progression. The liver's anatomical location, which may serve as a trafficking hub for CRC cells to spread to other organs, possibly underlies this finding[53]. By modeling large autopsy data sets in mCRC, Newton et al. highlighted that liver metastases could serve as tumor spreaders[54], and that there are multidirectional paths of tumor spread during progression[54,55]. Although we did not estimate transit probabilities from site to site, we speculate it is likely that early progression in liver metastases could expand systemic peripheral tolerance and promote more frequent and rapid progression in other organs. Our population-level analysis supports this speculation and shows that liver metastases were often associated with a more pronounced tumor spread in the body.

The primary therapeutic benefit of antibody-targeted therapies was to delay tumor progression and systemic relapses, without clear preferential effect on any organ-specific metastases. As such, antibody therapies did not seem to affect lesion progression sequences, and the fraction of patients with the first progression in the liver were largely comparable to chemotherapy alone. Unfortunately, in patients with multiple progressive metastases (> 4 progressive organs), the therapeutic benefit of antibody therapies is minimal, and more effective treatments remain sorely needed for advanced patients with extensive metastases.

Our study has limitations. First, the size of metastases was measured by radiological CT scans, but lesions in the bone are generally hard to assess, which could result in quantification bias and variability. Second, we also should note that patients in our dataset were from randomized control trials, who sometimes have different demographic characteristics with real-world patients. The average age of the patients in our dataset was 60.2, younger than the average age of diagnosis in real world patients[15]. Third, we pooled patients from multiple trials and these patients had very discrete surgical and treatment histories. Even though we controlled our analysis by considering these factors as covariates, direct trial-to-trial comparisons should be prevented. Machine learning has therefore identified distinct patterns for tumor progression, but future validation of these findings will be through acquisition and study of further independent datasets.

In conclusion, we quantified lesion response and progression heterogeneity by modeling the longitudinal size measurement of metastatic lesions. This study provided a broad characterization of phenotypic heterogeneity across metastatic lesions in mCRC, which could complement conventional molecular and cellular analyses to and promote a more comprehensive view of lesion-specific response heterogeneity and yield substantial implications for metastasis-targeting therapies.

## Methods

### Data

Multiple mCRC studies with longitudinal measurements of individual metastatic tumor information were included for the analyses. All datasets are accessible in Project Data Sphere, an open-access platform that aggregates cancer clinical trial data from biopharmaceutical companies, academic medical centers, and government organizations(https://www.projectdatasphere.org/). Patients under one of the following conditions were excluded: (1) no target lesion longitudinal measurements; (2) baseline tumor size measured more than 12 weeks before the treatment. Patients' demographics and survival information were collected if applicable. The size and anatomical site about target/non-target lesion and occurring time and anatomical sites of new lesions were all recorded and analyzed if any. Data from a Phase III study on panitumumab plus chemotherapy for mHNSCC were also collected from Project Data Sphere and were analyzed with same workflow for validation purpose. Data were processed in in R-4.1.0 and RStudio 2022.07.1 dplyr package. The clinical trials information was provided in Supplementary Data 1.

All study protocols were approved by institutional review boards at each participating center, including the clinical trial review boards from Amgen Inc., Pfizer Inc., Sanofi Inc., and AstraZeneca Inc. All patients have been provided written informed consent before study-related procedures were performed. All data sharing plans have been approved by the data sponsors.

## Lesion-specific tumor growth dynamics

The longest diameter was converted to volume assuming the ellipsoidal shape of tumor (1) and the ratio of the tumor long versus short axis as 1.31[56]. An empirical tumor growth model (2) was used to recapitulate lesion-specific tumor growth dynamics.

$$V = \frac{(long\ axis) \times (short\ axis)^2}{2} \tag{1}$$

$$V = V0 \cdot [F \cdot e^{Kg \cdot t} + (1 - F) \cdot e^{-Kd \cdot t}] \tag{2}$$

$V$ is the tumor volume, $V0$ is the tumor baseline volume, $t$ is the time. The model has three parameters for estimation: $F$ is the fraction of non-responding tumor cells, with $1-F$ as the response depth; $Kg$ is the progression rate and $Kd$ is the regression rate. We fitted the model for all target lesions simultaneously using the Non-Linear Mixed Effect (NLME) method in Monolix2020R1 Lixoft. Stochastic approximation expectation-maximization (SAEM) algorithm[57] was applied to search global optimum in the estimation. M3 method[58] was applied for quantifying size below the quantification of limit ($< 200\ mm^3$)[59]. In the NLME method, the model parameters are described in (3)-(5).

$$\ln(Kg^j) = \ln(\theta_{Kg}) + \eta_{Kg^j} \tag{3}$$

$$\ln(Kd^j) = \ln(\theta_{Kd}) + \eta_{Kd^j} \tag{4}$$

$$\mathrm{logit}(F^j) = \mathrm{logit}(\theta_F) + \eta_{F^j} \tag{5}$$

where $\theta$ is the population typical value, and $\eta$ is the random effect with a log-normal and logit-normal distribution describing the difference between individuals and population average for each lesion $j$. Proportional error model was assumed. The initial values of $Kg$, $Kd$ and $F$ were 0.01 day$^{-1}$, 0.01 day$^{-1}$, and 0.1 (unitless).

Of note, the volumetric conversion would make the thresholds for response and progression different from dimensional metrics. However, our model system assumes first-order dynamics of tumor regression and progression, and the response and progression sequence at the organ level remains unchanged regardless of the use of volumetric or dimension metrics.

## Tumor time to response and progression

Tumor growth dynamic parameters were further taken to predict the longitudinal profiles of response and progression for each target lesion. The longitudinal response and progression status for each target or non-target lesion were determined per RECIST V1.1[26]. Target lesion time to response (when the lesion volume decreases $\geq 20\%$ from baseline) and time to progression (when the lesion volume increases $\geq 30\%$ from tumor nadir or at least $200\ mm^3$ increase from nadir) were derived using tumor growth model with NLME-estimated parameters on the individual lesion level. Non-target lesions responded when partial response or complete response was firstly observed during the treatment and progressed when progressive disease appeared in tumor evaluation. The time to progression for new lesions were defined as the detection time.

## Cox proportional regression model

Cox proportional models were built to estimate lesion response and progression probabilities across organs and treatments in R-4.1.0 and RStudio 2022.07.1 coxme package. Inter-patient variability was adjusted in the Cox models as random effect. The potential covariate effects of treatment, line of therapy, age, gender, race, BMI, and surgical history were tested in mCRC patients and significant covariates ($p$-value $< 0.05$) were selected in the final model. For mHNSCC, we tested treatment, line of therapy, age, gender, race, BSA, and surgical/radiation history as potential covariates. Lesions without progression or response during the treatment were labeled as censored by the last day of that patient in the trial. New lesions were considered only in the progression hazard estimation.

## Progression pattern classification and prediction

We used the k-means clustering algorithm[32] to categorize all the patients based on their organ progression sequence in Spyder (Python 3.8) in Anaconda using the SCIKIT-LEARN 1.0.2 software package. Elbow method[33], Silhouette score[34], AIC and BIC[35], were applied to find optimal k. The Elbow method selects optimal k based on the inflexion point of the performance curve. The Silhouette method is based on the similarity of a data-point to its own identified cluster and dissimilarity to other clusters. The AIC and BIC provide complementary measures that balance model complexity and predictive performance. The optimal k was also selected to yield adequate separation of patient survival and interpretable features of lesion progression.

Gradient Boosting algorithm[36] was applied to build a progression pattern predictive model in Spyder (Python 3.8) in Anaconda using the SCIKIT-LEARN 1.0.2 software package. The research samples were randomly divided into training and testing sets at a ratio of 4:1, with rotation of the test dataset to implement 5-fold cross validation. The initial value of the hyperparameters used in this model was determined by parameter grid search, using 5-fold cross-validation and F1-score as a metric (Supplementary Table 4). The model outcome is the patient progression sequence classified in k-means clustering algorithm. Model inputs included patient clinical and demographic characteristics, as well as the baseline metastatic profiles, including the metastatic organs, metastatic numbers, metastatic target lesion baseline volume. Continuous predictors were normalized and categorical predictors were transformed to dummy variables using OneHotEncoder package softmax function. Performance index accuracy, precision, recall rate and area ROC curves were used to evaluate model performance.

## Statistics & reproducibility

Comparisons of continuous variables were performed using the two-tailed Mann–Whitney test or Kruskal–Wallis test. Multiple comparisons were adjusted by Dunn's test. PFS (defined as the start of therapies until RECIST-defined progression or death) and OS (defined as the start of therapies until patient death) among the groups were depicted using Kaplan–Meier curves and compared using log-rank tests. All the statistical tests were performed in GraphPad Prism 9. No statistical method was used to predetermine sample size. Data were included or excluded based on pre-established criteria. No randomization was involved in our analysis and the investigators were not blinded to allocation during data analyses and outcome assessments.

## Reporting summary

Further information on research design is available in the Nature Portfolio Reporting Summary linked to this article.

# Data availability

The raw clinical data that support the findings of this study are available in the Project Data Sphere, https://data.projectdatasphere.org/projectdatasphere/html/access. Access can be acquired on the Project

Data Sphere website. The processed data generated in the study are provided in the Source Data file with this paper. Source data are provided with this paper.

## Code availability

The modeling and machine learning algorithms codes were deposited at https://github.com/zhoujw14/Mapping-Metastasis.git.

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

## Acknowledgements
We thank Mr. Timothy Qi and Dr. Tyler Dunlap from University of North Carolina at Chapel Hill, Eshelman School of Pharmacy for providing valuable suggestions and edits for the manuscript. Funding Source: National Institute of Health, R35GM119661.

## Author contributions
Conceptualizations: J.Z., and Y.C.; methodology: J.Z., A.C., G.F., Q.L., and Y.C.; formal analysis: J.Z.; investigation: J.Z., Y.L., Q.L., and Y.C.; writing-original draft: J.Z., and Y.C.; writing-reviewing and editing: J.Z., A.C., G.F., Y.L., Q.L., and Y.C.; supervision: Y.C.

## Competing interests
The authors declare no competing interests.
