## [Peer review file · Nature Communications]

REVIEWER COMMENTS

Reviewer #1 (Remarks to the Author): Clinical expert in colorectal cancer metastasis

Dear Editor

Thank you for the opportunity to critically review the manuscript titled “Mapping Inpatient Response Heterogeneity and Lesion-specific Relapse Dynamics in Metastatic Colorectal Cancer”

Short summary

The authors have used Project Data Sphere to generate a large cohort of patients with metastatic colorectal cancer. They have analysed the inpatient heterogeneity by modelling the longitudinal size measurement of metastatic lesions and made interpretations on response and relapse heterogeneity between different organs. Among many findings the authors conclude that patients with the first relapse in the liver experience worse survival outcomes compared to patients with first relapse in other organs. By this the authors claim to provide substantial implications for metastasis-targeting therapy.

Major comments

- Even though the manuscript is well written it will be hard to understand for someone outside the field of machine learning and modeling.
- The study cohort is poorly described. Based on the studies that the present cohort is generated from, it can be assumed that the cohort itself is extremely heterogeneous with a mix of curatively resected patients (primary tumor and/or first site of metastatic lesion – most often in the liver) to late palliative patients who have progressed on first, second and maybe third line chemotherapy. If so, this will greatly impact the results and all interpretations made from this study. This must be clearly stated and considered before any conclusions can be drawn. As a clinician working with these patients daily, the experience is that most patients are diagnosed with liver metastases as first site of metastatic disease. The patients in this cohort that did not have metastases in the liver to start with, was that simply because they had undergone liver surgery before inclusion in any of the studies that constitute the cohort of the present study.
- The rationale of this study is too rudimentary.
- There is quite some data on radiological heterogeneity coupled to prognosis in mCRC, as the authors state in the introduction. This impacts on novelty.

- The selection bias introduced when creating a study cohort based on patients included in RCTs is not mentioned but is clearly stated by the mean age of the study cohort that is 10 years lower than normal as is the gender distribution that does not correspond to the normal gender distribution in mCRC.
- The term relapse, the frequent outcome of this study, is confusing as it indicates complete response (after surgery) and then tumour relapse. The use of the words recurrence and progression would instead make the text more understandable to a clinician.
- Could there be a systematic error from measuring 2D diameters (if that is what was measured)? I.e. do they assume regression to 50% if a lesion regressed from 4 to 2 cm? This would be relevant as this is wrong in terms of volume? Please correct me if I understood this wrong
- The discussion is very speculative and, in several instances, (e.g. discussing the effect of Panitumumab and line 256 “Metastatic lesions with lower fractions of resistant cells also had slower progression rates” assumes knowledge of tumor biology that is not actually available (we know nothing about “the fraction of resistant cells” of a given lesions; we may speculate).
- As most patients are diagnosed with liver metastases as the first metastatic site, could it be that those not having liver metastases in this cohort have undergone liver resection then progressed and included in the studies that later forms the study cohort of the present study.

Minor comments

- line 48, reference to “14% 5-years relative survival” cites what appears to be an editorial of some sort that does not seem to contain any reference for this statement. (cannot access the other reference)
- Intro does not give any clue as to the results and/or their applicability, which would help understanding the manuscript.
- Project Data Sphere is mentioned in beginning of results, but not explained in sufficient detail in the method section.
- In the results, it is not clearly stated which lesions were measured, how many (proportion of those found) were measured, which radiological method was used, etc.
- line 75: the rationale behind using a cohort of head and neck squamous cell carcinoma makes little sense and needs to be clearly stated and motivated. Why use a cohort of squamous cell carcinoma to validate a cohort with adenocarcinoma, two very distinct different cancer types
- line 81” recapitulated with a widely adopted growth model”: completely unclear what this means.
- similarly, “the fraction of non-responding cells” – what kind of radiologically identified cell would that be? Maybe trivial for someone in the field, but not for most readers

- “the model adequately recapitulated the longitudinal profiles of tumour radiographic” – adequate in relation to what? Measured against what? Judged how?

- line 90: lesions in the bone will give different results as they are much harder (at least very different from soft tissue lesions) to assess, particularly to call regression (rather than sclerosis).

- line 117, how were skin lesions assessed by radiology?

- line 128. “Patients with these metastases, particularly those with low-high phenotype, had much worse survival outcomes (OS median 378 days vs. 561 days, $p < 0.0001$)” – this is trivial (bad outcome for those with low response and high relapse), and it is unclear which group the 561 belong to?

- line 146: unclear that they switch back to mCRC

- line 115: “abdominal lesions”, what is included in that? Liver, spleen, peritoneum and abdominal lymph nodes are all abdominal organs...

- line 168: “Remarkably, despite comparable number of baseline metastases, patients whose first relapses were in the liver had a much worse prognosis than those whose first relapses were in lungs or other sites.”. “Number of metastases at baseline” is something that is never used in the clinic. What is interesting is what organ that is affected and how many different organs as that to some extent determines if the metastases are resectable.

It is also very well known that patients with “lung metastases only” do much better than metastases at other sites. That is why the presence of few and small lung metastases are not considered when assessing a patient for curative metastasectomy.

- line 248: “the other antibody panitumumab in our analyses did not significantly affect tumor regression rates either, suggesting its low direct cytotoxicity in patients” I believe the authors do not have the data to allow an analysis of such an effect of Panitumumab (as this would require an RCT).

“Relapse” in brain metastases (in 67) – should that be “progression”? Or was there complete response (must be extremely rare) in brain mets for 67 patients? How does that fit to response in only 23? (Figure 3 a/b)

True “Skin” metastases are very rare as a manifestation of mCRC. Do the authors mean cutaneous metastases?

Line 362: defining OS from start of therapies makes the results non-comparable with other studies. Either the start should be diagnosis of primary tumor or diagnosis of first metastatic lesion. If not, all sorts of bias are introduced. One could potentially compare someone who has undergone metastatic surgery and later treated with systemic therapy and then finally included in one of the mentioned trials to a patient diagnosed with mCRC and immediately included in one of the studies.

Same with PFS (defined as the start of therapies until RECIST-defined progression or death). The time with mCRC prior to inclusion in any of the studies must differ greatly.

Table 1, line 520. The mean age indicate that the cohort consists of highly selected patients as normally, the mean age is around 70 year. Same with the gender distribution, less females than normally – why so? This needs to be addressed.

70% have undergone previous surgery. This is crucial for the interpretation of the entire study. Curative resection of primary only? Or metastatic resection? Must be clearly stated in order for the reader to interpret the results.

Figures

Figure 1b, line 523. What are abdominal metastases? Skin lesions more common than brain metastases? That does not correlate accurately to the clinical experience. Do you mean cutaneous metastases?

Reviewer #2 (Remarks to the Author): Expert in colorectal cancer and computational genomics

Zhou et al. have modeled lesion-specific responses to systemic treatment across eight previously published clinical trials of metastatic colorectal cancer. Size measurements of multiple lesions and at multiple time points have been analyzed for each of 4,308 patients using a previously published method to model three different parameters, including the progression rate, regression rate and the fraction of non-responding cells. This is a comprehensive study revealing interesting features of organ-specific tumor growth dynamics during standard treatment of metastatic colorectal cancer, and should be of interest to the field. However, there are some issues that should be addressed.

1. The title, abstract and conclusion indicate that the study focuses on inpatient heterogeneity of treatment responses. With analysis of multiple lesions and multiple time points for each patient, data to analyze inpatient heterogeneity are available and this would be highly interesting. However, no direct

measures of inpatient heterogeneity have been described or analyzed in the manuscript, and this does not seem to be a focus of the study. Instead, lesion-specific data are summarized per metastatic organ and compared across patients. I recommend rephrasing the text to more accurately reflect the purpose of the study.

2. The clinical trials analyzed are a mix of studies of 1st and 2nd line treatment of metastatic colorectal cancer. The treatment line is a factor that needs to be accounted for in the analyses, considering its major impact on responses.

3. The performance of the tumor growth dynamics models is evaluated by comparisons with observed tumor volume data, showing correspondence across lesions and in inter-lesion variances. These models are likely to provide a more dynamic response evaluation than the criteria used in the clinical trials, and it would be interesting to also compare directly the modeled and observed responses and relapses (the conclusions from the trials at defined time points). This could provide both a more intuitive evaluation of the models, and specify the potential advantages of the modeling.

4. The response parameters analyzed for prognostic associations should be evaluated in multivariable survival models including the parameters listed in Table 1, for characterization of prognostic value in relation to known prognostic factors.

5. The number of metastatic organs should be considered in survival analyses of the four phenotypic features in supplementary fig 3. This would be particularly relevant to elucidate if the superior survival associated with metastases in high-low phenotype organs is because of the anatomical location of the metastases, or simply because the remaining patients have more extensive metastatic disease (involvement of more organs). It would also be interesting to see if results are similar in separate analyses of target lesions and non-target lesions.

6. The interpretation of the $k=5$ unsupervised clusters is somewhat confusing, in particular in survival analysis. There is presumably an overlap in characteristics among the groups, e.g. the “hetero-organ” group also has a substantial number of “liver-first” patients? To provide a clear analysis of the prognostic impact of the relapse sequence, it might be better to group the patients independent of the clustering analyses, that is, all patients with liver-first metastases versus the rest, etc. It is not surprising that patients with metastases to multiple organs have a poor survival, and this should rather be a stratification factor in prognostic analyses of the relapse sequence.

7. Analyses are stratified according to treatment regimens, but only for chemo versus chemo plus antibodies. It is relevant to also compare chemo regimens FOLFOX versus FOLFIRI.

8. Results are summarized per metastatic organ and compared across patients, but is there variation among lesions per organ/patient? It would be interesting to evaluate the inpatient intra-organ heterogeneity in treatment response and relapse where possible.

Minor comments:

1. Some relevant details are missing from the description of the patient data, e.g. an overview of the number of lesions and time points evaluated per patient, as well as treatment doses and cycles and imaging modalities across studies. This info would indicate if stratified analyses by study are needed.

Furthermore, the overview of clinical trials indicates that all studies were randomized phase III studies, but it appears that only one treatment arm was included for several. Please provide a description and justification of the patient selection.

2. Table 1:

- o It is stated that PFS and OS are reported in Table 1, but this information is missing. This would be helpful in the interpretation of the survival analyses.

- o Patient numbers for treatment add up to more than 4308.

3. The survival analyses shown in supplementary figure 6 are not in line with the description in the text. Is the labeling correct in parts b and c?

Reviewer #3 (Remarks to the Author): Expert in colorectal cancer epidemiology

The authors present a novel analysis with some intriguing findings further illustrating the heterogeneity of metastatic colorectal cancer. I have only a few comments, mostly regarding the presentation.

- Some of the lesion sites included in the analysis are modeled based on very few observations (e.g., skin, pancreas, bone), leading to questions as to the robustness of the model for these sites. The authors should consider focusing their analysis on the 8-10 most represented sites.

- The use of a validation set that includes patients with metastatic cancers of a different primary site should be better justified.

- The objectively measured outcomes should be more heavily weighted in the authors' presentation than the model-estimated parameters.

Reviewer #4 (Remarks to the Author): Expert in machine learning, statistics and bioinformatics

The reviewer's background is in machine learning and bioinformatics and consequently the comments below apply to these aspects of the paper, rather than the more general discussion of intrapatient response heterogeneity.

The paper was interesting to read, and well written. In terms of the data analysis and machine learning there were some shortfalls with the techniques used which need clarifying, and particularly, slightly sloppy descriptions of models: those familiar with the area would understand the approach used, though.

I. 158: 'Relapse sequence across organs predicts patient survival'. At this point the authors discuss the use of the Elbow method and the Silhouette method to find K, the number of proposed clusters in a K means clustering algorithm. The paper refers to Supplementary Figures 5A and 5B. Firstly, the caption below Figure 5 looks inadequate, simply saying the Elbow method and Silhouette method were used. I think it would help to provide some more detail here. Specifically, the Elbow method is a heuristic based on identifying the inflexion point of the curve. By sight this does look to be about 5. The Silhouette method is based on the similarity of a datapoint to its own identified cluster and dissimilarity to other proposed clusters. In this context it is not particularly clear from Figure 5B that K=5 is a suitable choice, at least better than other choices? Similar remarks apply in the area of I. 177 where Figure 5C and 5D are discussed.

Determining the optimal number of clusters in the data is something of a 'can of worms' issue since it involves avoiding fits to noise, the dual objectives of maximising intra-cluster similarities and inter-cluster dissimilarities, and possible variant methods, e.g. mixed membership models (data points can potentially overlap several clusters), may give a better model.

In short, I think the approach should be: (a) a variety of methods are used to determine K and an objective discussion is given of the concordance or discordance between these different methods, (b) you may have to run with a model which is not unambiguously justified (in terms of the best choice of K). In that case the model has to be explanatory and, in particular, justified on independent held-out data which validates the choice of K. For (a) I think it could be interesting to have a look at other clustering criteria, such as the Akaike information criterion (AIC) or Bayesian information criterion (BIC): do they also indicate K=5? If not, it is appropriate to objectively discuss the concordance and discordance between criteria and go with a model which would appear to work best according to be (b). Certainly, the K(=5) (I.161) isn't properly justified.

I. 202. 'Machine learning model predicts lesion relapse sequence'. Though I can't comment on the meaningfulness of the identified categories, gradient boosting is a good choice here and the ROC curves in Figure 8 indicate a reasonable predictive model.

I. 319. 'for stochastic approximation expectation maximisation': a citation is suitable here (there are various models of this type).

I. 343. 'Relapse pattern classification and prediction'. Some of the description here is a little unclear, though correct. First sentence might be better phrased 'We used the K means clustering algorithm, from machine learning, to categorise all the patients based on their organ relapse.' ['categorise' sounds better than 'classify' since the latter is a bit associated with supervised learning in the context of machine learning]. In the next sentence they should provide citations after Elbow method and Silhouette score. In this paragraph an issue which was not covered was if data input attributes coming into these models were standardised or normalised in some way. If attributes are on very different scales this would affect the performance of a clustering or machine learning algorithm. I'm not sure if the authors applied their

own standardisation or, quite possibly, data standardisation was incorporated within the algorithms and packages which they used? However, this is an issue which should be at least briefly mentioned.

L. 350. 'into a training and testing groups at a ratio of 4:1'. In the follow on sentence is it is apparent that the authors have simply used five-fold cross validation, however, this sentence does not exclude the possibility that the algorithm was trained on 80% of the data and a single held out 20% of the data. Perhaps a better wording might be 'the research samples were randomly divided into training test set according to the ratio of 4:1, with rotation of the test set data to implement 5-fold cross validation.'

L. 353. K means algorithm-> K means clustering algorithm? The next sentence begins 'Model predictors included patient clinical and demographic characteristics' : it is not clear to the reader from this phrasing if you are referring to the inputs to the model (the attributes or features used) or the outputs from the model, this should be clearly stated.

Manuscript NCOMMS-22-09853

Title: Mapping Lesion-Specific Response and Progression Dynamics and Inter-Organ Variability in Metastatic Colorectal Cancer

RESPONSES TO CRITIQUES

We would like to thank both reviewers as well as the editor for their careful reading of our paper and their thoughtful suggestions and comments. We have now addressed all comments in a revised version of the manuscript. Please find below a detailed point-by-point response to all issues raised, in which all major revisions are highlighted by **bold font** and deleted text shown with strikethroughs (~~deleted text~~) in the copy of the submission (manuscript with marked revisions). **The Page and Line # are highlighted in red.**

Response to Reviewer 1: Page 1- 16

Response to Reviewer 2: Page 17-26

Response to Reviewer 3: Page 27-28

Response to Reviewer 4: Page 29-33

REVIEWER COMMENTS

Reviewer #1

Thank you for the opportunity to critically review the manuscript titled “Mapping Inpatient Response Heterogeneity and Lesion-specific Relapse Dynamics in Metastatic Colorectal Cancer”

Short summery

The authors have used Project Data Sphere to generate a large cohort of patients with metastatic colorectal cancer. They have analysed the inpatient heterogeneity by modelling the longitudinal size measurement of metastatic lesions and made interpretations on response and relapse heterogeneity between different organs. Among many findings the authors conclude that patients with the first relapse in the liver experience worse survival outcomes compared to patients with first relapse in other organs. By this the authors claim to provide substantial implications for metastasis-targeting therapy.

Major comments

1. Even though the manuscript is well written it will be hard to understand for someone outside the field of machine learning and modeling.

Response: We thank the reviewer for pointing this out. We made the following clarifications in the Introduction, Method, and Results to make it easier to understand for readers outside the field of machine learning and modeling.

Revision: We rearranged **the Introduction** and added some contents for an ease understanding of our objectives and approaches.

In the **Introduction Page 3, Line 46-56**: The local microenvironment selects tumor phenotypes in response to treatment, leading to heterogeneity across anatomically distinct lesions in terms of response and progression dynamics. Characterizing their phenotypic differences (divergence) or similarities (convergence) could yield insights into tumor ecological features and systemic resistance. **The spatiotemporal patterns of response and progression at the lesion or organ level could not only be informative to prognosis, but also could enrich our knowledge of metastasis-to-metastasis interactions and the systemic consequence of regional progression.** ~~This study sought to investigate spatiotemporal response heterogeneity through mapping lesion-specific response and progression dynamics in metastatic CRC (mCRC).~~

In the **Results Page 5, Line 94-96**: There are three dynamic parameters in the model: ~~are~~ the regression (**tumor-killing**) rate Kd , the fraction of non-responding (**or resistant**) cells F , and the progression (**or regrowth**) rate Kg .

In the **Discussion Page 16, Line 342-346**: Although we did not estimate transit probabilities from site to site, we speculate it is likely that early **relapse progression** in liver metastases could ~~lead to more resistant cells spreading throughout the body and cause~~ **expand systemic peripheral tolerance and promote more frequent subsequent relapse and rapid progression in other organs.**

In the **Methods**.

We removed some mathematical contents and added more descriptions to the **Methods Page 20, Line 429-435**: Elbow method³³, and Silhouette score³⁴, **AIC and BIC**³⁴, were applied to find the optimal k . **The Elbow method selects optimal k based on the inflexion point of the performance curve. The Silhouette method is based on the similarity of a data-point to its own identified cluster and dissimilarity to other clusters. The AIC and BIC provide complementary measures that balance model complexity and predictive performance. The optimal k was also selected to yield adequate separation of patient survival and interpretable features of lesion progression.** ~~The relapse patterns of patients clustered with different k were compared to help determine the choice of k in the final classification.~~

More references were added:

References Page 25, Line 522-524: 33. Syakur, M. A., Khotimah, B. K., Rochman, E. M. S. & Satoto, B. D. Integration k-means clustering method and elbow method for identification of the best customer profile cluster. in IOP conference series: materials science and engineering vol. 336 12017 (IOP Publishing, 2018).

References Page 25, Line 525-527: 34. Shahapure, K. R. & Nicholas, C. Cluster quality analysis using silhouette score. in 2020 IEEE 7th International Conference on Data Science and Advanced Analytics (DSAA) 747–748 (IEEE, 2020).

References Page 25, Line 528-529: 35. Burnham, K. P. & Anderson, D. R. Multimodel inference: understanding AIC and BIC in model selection. *Sociol. Methods Res.* 33, 261–304 (2004).

We added citation to the **Methods Page 19, Line 394**: Stochastic approximation expectation-maximization (SAEM) algorithm⁵⁷ was applied to search global optimum in the estimation.

Here citation 57 were added to explain SAEM algorithm in the **References Page 27, Line 577-578**: Savic, R. & Lavielle, M. Performance in population models for count data, part II: a new SAEM algorithm. *J. Pharmacokinet. Pharmacodyn.* 36, 367–379 (2009).

In addition, in the Methods:

Page 20, Line 427: We used the k-means ~~machine learning~~ **clustering** algorithm to ~~classify~~ **categorize** all the patients...

We added how we standardized variables in Gradient Boosting model in **Page 21, Line 444-446:** Continuous predictors were normalized and categorical predictors were transformed to dummy variables **using “OneHotEncoder” package “softmax” function.**

Page 21, Line 438-439: The research samples were randomly split into a-training and testing ~~groups-sets~~ at a ratio of 4:1, **with rotation of the test dataset to implement 5-fold cross validation.**

Page 21, Line 441-442: The model outcome is the patient ~~relapse~~ **progression** sequence classified in k-means **clustering** algorithm. Model ~~predictors~~ **inputs** included patient clinical and demographic characteristics...

We added citation 32 to explain the unsupervised k-means clustering model in the **Results Page 9, Line 194:** We built a k-means unsupervised clustering model³²... and the **Methods Page 20, Line 427:** We used the k-means ~~machine learning~~ **clustering** algorithm³²... Please find citation 32 in **References Page 25, Line 520-521:** 32. Sinaga, K. P. & Yang, M.-S. Unsupervised K-means clustering algorithm. IEEE access 8, 80716–80727 (2020).

We added citation 36 to explain gradient boosting model in the **Method Page 21, Line 436:** Gradient Boosting algorithm³⁶ was applied to build a relapse pattern predictive model... Please find citation 36 in the **References Page 25, Line 530:** 36. Natekin, A. & Knoll, A. Gradient boosting machines, a tutorial. Front. Neurobot. 7, 21 (2013).

2. The study cohort is poorly described. Based on the studies that the present cohort is generated from, it can be assumed that the cohort itself is extremely heterogenous with a mix of curatively resected patients (primary tumor and/or first site of metastatic lesion – most often in the liver) to late palliative patients who have progressed on first, second and maybe third line chemotherapy. If so, this will greatly impact the results and all interpretations made from this study. This must be clearly stated and considered before any conclusions can be drawn. As a clinician working with these patients daily, the experience is that most patients are diagnosed with liver metastases as first site of metastatic disease. The patients in this cohort that did not have metastases in the liver to start with, was that simply because they had undergone liver surgery before inclusion in any of the studies that constitute the cohort of the present study.

Response: We agree that many factors mentioned by the reviewer could confound the results and our conclusion. We provided more clarification on patient surgical and treatment history in **Supplementary Table 5.** And in our model, we evaluated these factors as **potential covariates and included the select significant covariates in the final model** to correct the potential bias.

Specifically, in our dataset, about 69.5% patients (n=2,993) had curative surgery, from which about 6% patients (n=261) had resection surgery of liver metastases; 0.85% patients (n=37) had lung lesion resection; 4.8% patients (n=206) had resection of lesions in local and distal lymph nodes. In the dataset, 2,216 patients (patients in NCT00364013, NCT00115765, NCT00272051, NCT00305188, and NCT00457691) did not receive any prior chemotherapy. Patients in NCT00339183 were allowed to have adjuvant radio-sensitizing chemotherapy but it had to be over 6 months before study entry. Patients in NCT00561470 were allowed have 1 prior chemotherapeutic regimen, including oxaliplatin containing regimen. In summary, 2,946 patients (68.3%) in our dataset entered the study as their first-line therapy and the confounding effect of treatment history was evaluated.

Revision: More trial and patient information were added into **Supplementary Table 5**, including patient surgical and treatment history. In the **Methods Page 18, Line 379-380: The clinical trials information was provided in Supplementary Table 5.**

Moreover, we did some further analyses to evaluate the confounding effect of surgical and treatment history. Specifically, we tested surgical and treatment history as potential covariates and included the significant covariates in the final model. We next compared organ-specific hazard distributions between patients w/ and w/o these covariates.

Supplementary Figure 4 in the **Supplementary Page 9, Line 63-65** shows the covariate effects:

Supplementary Fig. 4. Covariate effects on lesion response and progression. **a** and **b** rank the hazard ratios with 95% confidence interval of each potential covariate.

We also acknowledged that our study pooled patients from multiple trials and the included patients had very distinct surgical and treatment history. In the **Discussion Page 17, Line 359-362: Our study has limitations. ... Third, we pooled patients from multiple trials and these patients had very discrete surgical and treatment histories. Even though we controlled our analysis by considering these factors as covariates, direct trial-to-trial comparisons should be prevented.**

3. The rationale of this study is too rudimentary.

Response: We added the following contents to rationalize our study.

Revision: In the **Introduction Page 3, Line 52-56** we added: **The spatiotemporal patterns of response and progression at the lesion or organ level could not only be informative to prognosis, but also could enrich our knowledge of metastasis-to-metastasis interactions and the systemic consequence of regional progression. This study sought to investigate spatiotemporal response heterogeneity through mapping lesion-specific response and progression dynamics in metastatic CRC (mCRC).**

And **Introduction Page 4, Line 72-75:** Last, we applied a machine learning approach to analyze the progression sequence across lesions and its relevance to long-term patient survival. **The spatiotemporal patterns of response and progression at the organ level could infer tumor evolution in space and time, affording more biological hypotheses.”**

4. There is quite some data on radiological heterogeneity coupled to prognosis in mCRC, as the authors state in the introduction. This impacts on novelty.

Response: Literature studies have reported the prognostic values of tumor heterogeneity in radiological size and metastatic site, and we have cited most of these reports, including our previous one published in Cancer Res (Reference #18 in manuscript). To our knowledge, this study is the first one to apply the dynamic parameters in tumor growth and progression at the organ level to inform patient prognosis. More importantly, we found the spatiotemporal patterns of progression at the organ level is highly informative to patient long-term survival, which has many clinical implications.

Revision: Introduction Page 4, Line 68-76: **To map the lesion-level response and relapse progression patterns in mCRC, we first applied a mathematical model to capture tumor growth dynamics in 4,308 mCRC patients. Next, individual lesion-specific response and relapse progression probabilities were mapped to predict their phenotypic divergence and convergence across anatomical sites. Last, we applied a machine learning approach to analyze the relapse progression sequence across lesions organs and its relevance to long-term patient survival. The spatiotemporal patterns of response and progression at the organ level could infer tumor evolution in space and time, affording more biological hypotheses.** Our study provides insights into ~~intrapatient~~ **lesion-specific** phenotypic heterogeneity in mCRC and yields substantial implications for designing metastasis-specific therapeutics.

5. The selection bias introduced when creating a study cohort based on patients included in RCTs is not mentioned but is clearly stated by the mean age of the study cohort that is 10 years lower than normal as is the gender distribution that does not correspond to the normal gender distribution in mCRC.

Response: This is a great point! Demographic discrepancy between RCT and real-world patients has become a major issue for clinical drug evaluations. We compared patient demographic characteristics between our dataset and Cancer Statistics (Reference 15 in manuscript). The gender distribution is similar: 58.9% male in our dataset vs. 53.4% in Cancer Statistics. The patient age in our dataset is around 60.2 ± 10.8 (Mean \pm SD) and the Colorectal Cancer Statistics shows the average age of diagnosis is 68 for men for colon cancer and 63 for rectal cancer. Patients in RCTs trials were generally younger than real world patients. Of note, most RCTs included in our dataset were conducted around early 2000, which might explain the discrepancy of patient characteristics.

Revision: Discussion Page 17, Line 355-358: **Our study has limitations. ... Second, we also should note that patients in our dataset were from randomized control trials, who sometimes have different demographic characteristics with real-world patients. The average age of the patients in our dataset was 60.2, younger than the average age of diagnosis in real-world patients¹⁵.**

6. The term relapse, the frequent outcome of this study, is confusing as it indicates complete response (after surgery) and then tumour relapse. The use of the words recurrence and progression would instead make the text more understandable to a clinician.

Response: An excellent point! We changed most “relapse” terms to either “recurrence” or “progression”:

Title Page 1, Line 1-2: Mapping ~~Intrapatent Response Heterogeneity and Lesion~~ **Lesion-Specific Response and Relapse Progression Dynamics and Inter-Organ Variability** in Metastatic Colorectal Cancer

In the **Abstract:**

Page 2, Line 23-24: ...such as lesion dynamics in growth, response, and ~~relapse~~ **relapse progression** during treatment, ...

Page 2, Line 25-30: Despite significant differences in their response and ~~relapse~~ **relapse progression** dynamics, ... Importantly, we found that organ-level ~~relapse~~ **relapse progression** sequence was closely associated with patient survival, and that patients with the first ~~relapses~~ **relapses progression** in the liver often had worse survival. ... our study provides insights into ~~intrapatient~~ **lesion-specific response and progression** heterogeneity...

In the **Introduction**

Page 3, Line 37: Lesion-level response and ~~relapse~~ **relapse progression** heterogeneity are common in many cancer types, ...

Page 3, Line 44: ...growth, response, and ~~relapse~~ **relapse recurrence** during treatment, ...

Page 4, Line 68-72: To map the lesion-level response and ~~relapse~~ **relapse progression** patterns in mCRC, ... Next, individual lesion-specific response and ~~relapse~~ **relapse progression** probabilities were mapped to predict ... Last, we applied a machine learning approach to analyze the ~~relapse~~ **relapse progression** sequence across lesions ...

In the **Results:**

Page 5, Line 80: To evaluate ~~lesion-level metastatic~~ response and ~~relapse~~ **relapse progression** dynamics in mCRC, ...

Page 6, Line 122: Response and ~~relapse~~ **relapse progression** dynamics suggest phenotypic convergence on organ level

Page 6-7, Line 123-125: The tumor growth model predicted the longitudinal profiles of response and ~~relapse~~ **relapse progression** for each target lesion. Response and ~~relapse~~ **relapse progression** times were then derived as the duration from the start of treatment to the time of response or ~~relapse~~ **relapse progression** per RECIST v1.1, respectively.

Page 7, Line 129-137: We integrated the response time for both target and non-target lesions and the ~~relapse~~ **relapse progression** time for all lesions, including the new ones, into random effect Cox proportional models. The Cox model predicted the relative probabilities of lesion response or ~~relapse~~ **relapse progression** at the organ level... in the ~~Cox regression~~ **relapse progression** model. ...

Page 7, Line 140-141: ...response and ~~relapse~~ **relapse progression** characteristics. The hazard ratios for the response and ~~relapse~~ **relapse progression** ...

Page 7, Line 145-147: The probability of ~~relapse~~ **relapse progression** also differed greatly across anatomical sites (Fig. 3b). The metastatic lesions with the highest likelihood of ~~relapse~~ **relapse progression** were those in the brain/CNS, GR system, and liver, ...

Page 8, Line 149: We then integrated organ-specific response and ~~relapse~~ **relapse progression** probabilities...

Page 8, Line 151: ...specific response and **relapse progression** probabilities...

Page 8, Line 153-158: ...associated lesions' likelihood of response and **relapse progression**. Notably, bone and brain lesions had low response and high **relapse progression** probabilities ... both response and **relapse progression** ... lung and LN showed high response and low **relapse progression** probabilities...

Page 8, Line 162: ... most metastatic lesions with high **relapse progression** probabilities...

Page 8-9, Line 173-179: The organ-specific hazard ratios for **relapse progression** ... metastases in the liver, bone, and brain also showed high **relapse progression** potential, ... could be key modulators to the mechanisms underlying the probabilities of tumor response and **relapse progression**.

Page 9, Line 185-191: Notably, the primary therapeutic benefit of antibody targeted therapies was to decrease **relapse progression** potential ... Taken together, antibody targeted therapies had the primary effect of decreasing lesion **relapse progression** probability but had limited influence on the lesion response probability. Interestingly, high-**relapse progression** organs in Fig. 3c also had high **relapse progression** probability ...

Page 9, Line 193: **Relapse Progression** sequence across organs predicts patient survival

Page 9, Line 195: We built a k-means unsupervised clustering model to cluster patients based on their organ-level lesion **relapse progression** sequence...

Page 10, Line 201-203: ...identified with distinct patterns of organ-specific **relapse progression** sequences and were stratified by **relapsing progressive** organ number and first-**relapsing progressive** organ.

Page 10, Line 208-217: Organ-level **relapse progression** sequence is significantly correlated ... As expected, patients with multiple organ **relapses progression** had worse survival than patients with only one organ **relapse progression** ... Remarkably, despite comparable number of baseline metastases, patients whose first **relapses tumor progression** were in the liver had a much worse prognosis than those whose first **relapses progression** were in lungs or other sites ... Patients with **relapse progression** first in the liver had faster subsequent **relapses progression** than patients whose **relapses progression** occurred in lungs or other sites, suggesting that **relapsing progressive** lesions in the liver have high systemic ...

Page 10, Line 218-223: Next, ... to further investigate **relapse progression** patterns in patients with extensive metastases **and relapses progression**. ... one distinctive feature among these clusters was the **relapse progression** order of liver lesions ... patients with first or second **relapse progression** occurring in the liver had worse survival than those with early **relapses progression** occurring in other organs...

Page 11, Line 226: Targeted antibody therapies minimally influence lesion **relapse progression** sequence

Page 11, Line 227-237: We compared the **relapse progression** sequence ... In patients with Liver-First, Lung-First or Other-First **relapse progression** patterns, ... However, neither the proportion of patients with each **relapsing progressive** pattern (Fig. 5b) nor the sequence of **relapse progression** across metastatic organs were significantly different (Fig. 5c-5e). **Relapses Tumor progression** in the GR and pancreas ... patients under antibody targeted therapies had significantly slower first and second **relapses progression**, but had non-significant difference in the third or later **relapses progression** (Fig. 5f-5g). The average **relapse progression** times were much longer in combination therapy compared to chemotherapy alone.

Page 11, Line 241-246: Patients' ~~relapse~~ **relapse progression** patterns and lesion ~~relapse~~ **relapse progression** time were largely comparable, especially for those who had early liver lesion ~~relapse~~ **relapse progression** ... antibody targeted therapies did not influence lesion ~~relapse~~ **relapse progression** sequence. Overall, the primary therapeutic benefit of antibody targeted therapies was to delay ~~relapse~~ **relapse progression** ...

Page 11, Line 247: Machine learning model predicts lesion ~~relapse~~ **relapse progression** sequence

Page 12, Line 248: In order to predict patient ~~relapse~~ **relapse progression** sequence...

Page 12, Line 254: ...accurately predict the ~~relapse~~ **relapse progression** sequences of the latter three groups...

In the **Discussion**

Page 13, Line 261: ...response and ~~relapse~~ **relapse progression** dynamics...

Page 13, Line 265-273: ... Second, ...congruent response and ~~relapse~~ **relapse progression** dynamics, ... high response and high ~~relapse~~ **relapse progression** probabilities ...high ~~relapse~~ **relapse progression** probabilities ... organ-level ~~relapse~~ **relapse progression** sequence was closely associated with patient survival, and patients with the first ~~relapse~~ **relapse progression** in the liver had worse survival outcomes compared to patients with first ~~relapse~~ **relapse progression** in other sites.

Page 14, Line 289-294: Lesion ~~relapse~~ **relapse progression** time, **i.e., lead time to progression**, was more closely associated with ~~determined by~~ the progression rates...

Page 14, Line 300: ...more apparent after ~~relapse~~ **relapse progression**, ...

Page 15, Line 310-315: Metastatic lesions with a lower ~~fractions of resistant cells~~ **fraction of non-responding tumor cells (F)** also had a slower lesion progression rates. **The small fraction of resistant tumor cells prior to treatment implies their low fitness compared to sensitive populations, which led to low tumor regrowth rates** ~~suggesting consistent fitness of resistant cells before treatment and after~~ ~~relapse~~ **relapse progression**.

Page 15-16, Line 326-335: On the other hand, organ-specific ~~relapse~~ **relapse progression** probabilities ... Metastatic lesions with higher ~~relapse~~ **relapse progression** potentials ... Interestingly, high ~~relapses~~ **relapses progression** in these organs ... Higher containment of tumor ~~relapses~~ **relapses progression** in immunocompetent organs ... Patients with highly ~~relapsing~~ **relapsing progressive** lesions, ...

Page 16, Line 336-345: Tumor ~~relapse~~ **relapse progression** is a serious impediment to cancer treatment, but organ-level ~~relapse~~ **relapse progression** patterns remain poorly characterized. We found that early ~~relapses~~ **relapses progression** in the liver, ... rapid subsequent ~~relapses~~ **relapses progression**.

Page 16, Line 343-345: ...we speculate it is likely that early ~~relapses~~ **relapses progression** in liver metastases ... more frequent **and rapid** ~~subsequent~~ ~~relapses~~ **relapses progression**.

Page 16, Line 349-353: As such, antibody therapies did not affect ~~relapse~~ **relapse progression** sequences, ... the first ~~relapse~~ **relapse progression** in the liver ... patients with multiple ~~relapsed~~ **relapsed progressive** metastases (> 4 ~~relapsed~~ **relapsed progressive** organs), ...

In the **Methods**:

Page 19, Line 407: Tumor response and ~~relapse~~ **relapse progression** times

Page 19-20, Line 408-416: Tumor growth dynamic ... response and ~~relapses~~ **relapses progression** for each target lesions. The longitudinal response and ~~relapse~~ **relapse progression** status ... and ~~relapse~~ **relapse progression** time ...

was firstly observed during the treatment and ~~relapsed~~ **progressed** when “progressive disease” appeared in tumor evaluation. The ~~relapse~~ **progression** time for new lesions were defined as the detection time.

Page 20, Line 418-425: Cox proportional models were built to estimate lesion response and ~~relapse~~ **progression** probabilities ... Lesions without ~~relapse~~ **progression** or response ... only in the ~~relapse~~ **progression** hazard estimation.

Page 20, Line 426: ~~Relapse~~ **Progression** pattern classification and prediction

Page 20, Line 428: ...on their organ ~~relapse~~ **progression** sequence...

Page 21, Line 436: Gradient Boosting algorithm was applied to build a ~~relapse~~ **progression** pattern predictive model...

Page 21, Line 441: The model outcome is the patient ~~relapse~~ **progression** sequence...

In the **Figures**

Page 35-37, Line 622-629: The titles of Fig. 3b and Fig. 3f were changed from “Relapse” to “**Progression**”. The X-axis of Fig. 3c and Fig. 3d were changed from “Relapse probability” to “**Progression probability**”. **Legend:** Fig. 3 Organ-level tumor response and ~~relapse~~ **progression** ... a and b rank the hazard ratio estimates ... lesion response and ~~relapse~~ **progression** in colorectal cancer patients. c and d are the ... response and ~~relapse~~ **progression** hazard ratios in ... e and f are response and ~~relapse~~ **progression** hazard ratio ...

Page 38-39, Line 633-641: The Y-axis label of Fig. 4c were changed from “Time to relapse (days)” to “**Time to progression (days)**”. **Legend:** Fig. 4 Patient ~~relapse~~ **progression** ... a. Patients were clustered into five groups based on their lesion ~~relapse~~ **progression** sequence. The column labels are the ~~relapse~~ **progression** sequence ... c. The mean and standard deviation of the first lesion ~~relapse~~ **progression** time (1st), time between first and second ~~relapse~~ **progression** (2nd-1st), time between second and third ~~relapse~~ **progression** (3rd-2nd), time between third and fourth ~~relapse~~ **progression** (4th-3rd), and the average ~~relapse~~ **progression** time ...

Page 40-42, Line 642-652: The Y-axis label of Fig. 5f, 5g, and 5h were changed from “Time to relapse (days)” to “**Time to progression (days)**”. **Legend:** Fig. 5. Targeted therapy decreases average time to ~~relapse~~ **progression** but has minimal effect on ~~relapse~~ **progression** sequence. ... c, d, and e are patient ~~relapse~~ **progression** ... f, g, and h are the mean and standard deviation of the first lesion ~~relapse~~ **progression** time (1st), time between first and second ~~relapse~~ **progression** (2nd-1st), time between second and third ~~relapse~~ **progression** (3rd-2nd), time between third and fourth ~~relapse~~ **progression** (4th-3rd), and the average ~~relapse~~ **progression** ...

In the **Supplementary:**

Supplementary Page 5, Line 35-37: Supplementary Table 4. The hyperparameters of the gradient boosting model to predict patient ~~relapse~~ **progression** sequence.

Supplementary Page 11, Line 73-78: The title of Supplementary Fig. 6d were changed from “Relapse” to “**Progression**”. **Legend:** ... with p-values of lesion response and ~~relapse~~ **progression** by organs.

Supplementary Page 12, Line 80-84: ... method were applied to find optimal k in ~~relapse~~ **progression** patterns classification... were applied to find optimal k in ~~relapse~~ **progression** patterns...

Supplementary Page 13, Line 87-94: The Y-axis label of Supplementary Fig. 8c were changed from “Time to relapse (days)” to “**Time to progression (days)**”. **Legend:** Supplementary Fig. 8 Liver ~~relapse~~ **progression** sequence ... a. Hetero-Organ patients were clustered into four groups by liver ~~relapse~~ **progression** sequence. b. Kaplan-Meier curves of patient overall survival by liver ~~relapse~~ **progression** sequence. c. The mean and standard deviation of the first lesion ~~relapse~~ **progression** time (1st), time between first and second ~~relapse~~ **progression** (2nd-1st), time between second and third ~~relapse~~ **progression** (3rd-2nd), time between third and fourth ~~relapse~~ **progression** (4th-3rd), and the average ~~relapse~~ **progression** time ...

Supplementary Page 14, Line 99-108: The Y-axis label of Supplementary Fig. 9f, 9g, and 9h were changed from “Time to relapse (days)” to “**Time to progression (days)**”. **Legend:** Supplementary Fig. 9 Targeted therapy has minimal effect on **progression** sequence order. c, d, and e Patient ~~relapse~~ **progression** sequences stratified by treatments. f, g, and h The mean and standard deviation of the first lesion ~~relapse~~ **progression** time (1st), time between first and second ~~relapse~~ **progression** (2nd-1st), time between second and third ~~relapse~~ **progression** (3rd-2nd), time between third and fourth ~~relapse~~ **progression** (4th-3rd), and the average ~~relapse~~ **progression** time ...

Supplementary Page 15, Line 109-115: Supplementary Fig. 10 ~~Relapse~~ **Progression** sequence prediction model performance.

7. Could there be a systematic error from measuring 2D diameters (if that is what was measured)? I.e. do they assume regression to 50% if a lesion regressed from 4 to 2 cm? This would be relevant as this is wrong in terms of volume? Please correct me if I understood this wrong

Response: This is a great point. Volumetric conversion assuming ellipsoidal shape of tumor is a common approach when quantifying tumor growth dynamics, considering volumetric metrics is a more suitable proxy to tumor mass. We used equation 1 for this conversion. The reviewer is correct that volumetric metrics shifts the percentage of tumor change compared to the dimension metrics (i.e., RECIST 1.1). For instance, a 50% of volume increase results in only ~ 15% increase of tumor dimension. The volumetric conversion would make the thresholds for response and progression different from dimensional metrics per RECIST 1.1. We’re well aware of the difference. However, our model system assumes first-order dynamics of tumor regression and progression, and the response and progression sequence at the organ level remains unchanged regardless of the use of volumetric or dimension metrics. For instance, lesions in the liver often progress at a higher rate than lesions in the lymph nodes, and the order of progression would be the same under both metrics (volumetric or dimensional).

$$V = \frac{(long\ axis) \times (short\ axis)^2}{2} \text{ (Equation 1)}$$

Revision: In the **Methods Page 19, Line 403-406:** **The volumetric conversion would make the thresholds for response and progression different from dimensional metrics. However, our model system assumes first-order dynamics of tumor regression and progression, and the response and progression sequence at the organ level remains unchanged regardless of the use of volumetric or dimension metrics.**

In the **Methods Page 19, Line 410-412:** Target lesion response time (when the lesion size **volume** decreases $\geq 20\%$ from baseline) and ~~relapse~~ **progression** time (when the lesion size **volume** increases $\geq 30\%$ from tumor nadir or at least 200 mm³ increase from nadir)...

8. The discussion is very speculative and, in several instances, (e.g. discussing the effect of Panitumumab and line 256 “Metastatic lesions with lower fractions of resistant cells also had slower progression rates” assumes knowledge of tumor biology that is not actually available (we know nothing about “the fraction of resistant cells” of a given lesions; we may speculate).

Response: We removed some speculative contents about the cytotoxic effect of panitumumab in the Discussions section and also reworded some speculative contents. In addition, we added several additional references to support our discussion.

Revision: Discussion session Page 14-15, Line 303-312: Interestingly, despite the broad evidence of its antibody dependent cellular cytotoxicity (ADCC)⁴³ or complement dependent cytotoxicity in vitro systems⁴⁴, the other antibody panitumumab in our analyses did not significantly affect tumor regression rates either. In fact, the magnitude of the ADCC elicited by EGFR targeting antibodies in patient remains hard to define, especially considering the restricted and highly varying infiltrations of effector cells in tumor beds^{45,46}. Panitumumab (IgG2), compared to another EGFR targeting antibody cetuximab (IgG1) showed reduced ADCC dependent therapeutic effect, probably related to the reduced avidity of IgG2 for CD16, as compared to IgG1^{47,48}. Unfortunately, our analyses did not include patients under cetuximab treatment, precluding direct comparison.

Page 15, Line 311-314: Metastatic lesions with a lower fraction of resistant cells **fraction of non-responding tumor cells (F)** also had a slower lesion progression rates,. **The small fraction of non-responding (resistant) tumor cells prior to treatment implies their low fitness compared to responding (sensitive) populations, which led to low tumor regrowth rates suggesting consistent fitness of resistant cells before treatment and after relapse progression.**

9. As most patients are diagnosed with liver metastases as the first metastatic site, could it be that those not having liver metastases in this cohort have undergone liver resection then progressed and included in the studies that later forms the study cohort of the present study.

Response: In our study, about 6% patients (n=261) had resection surgery of liver metastases before their entry to clinical trials. We performed further analysis and did not find these patients with resection of liver metastases have any particularly different response and progression patterns compared to those who did not have the surgery.

For more balanced analysis, we excluded patients without prior surgery history and re-analyzed the progression patterns (see the heatmap figure below). The progression patterns were quite similar with Figure 4, suggesting the minimal impact of surgical history on relapse sequence.

We also compared the survival in patients with prior surgery history:

From the analyses above, we can tell that our conclusions would not change when we excluded those patients without prior surgery history.

Minor comments

1. line 48, reference to “14% 5-years relative survival” cites what appears to be an editorial of some sort that does not seem to contain any reference for this statement. (cannot access the other reference)

Revision: This information could be found in the Reference 15: Biller, L. H. & Schrag, D. Diagnosis and treatment of metastatic colorectal cancer: a review. JAMA 325, 669–685 (2021).

2. Intro does not give any clue as to the results and/or their applicability, which would help understanding the manuscript.

Response: We agree with the reviewer. Therefore, we highlighted two major applications of our work in the Introduction: 1) prediction of patient prognosis based on metastatic sites and early response to treatments, which would afford rationale of metastasis-specific treatments; 2) hypothesis-generating about tumor resistance and evolution in space and time. The anatomical chart of lesion progression has strong implication about tumor heterogeneity across organs and how it relates to patient response at the system level. The sequence of progression at the organ level suggests potential lesion-lesion interactions, which implicates the systemic impact of regional lesions and potential tumor metastasis across lesions.

Revision: Introduction Page 4, Line 68-76: **In this study, to map the lesion-level response and relapse progression patterns in mCRC, we first applied a mathematical model to capture tumor growth dynamics in 4,308 mCRC patients. Next, individual lesion-specific response and relapse progression probabilities were mapped to predict their phenotypic divergence and convergence across anatomical sites. Last, we applied a machine learning approach to analyze the relapse progression sequence across lesions organs and its relevance to long-term patient survival. The spatiotemporal patterns of response and progression at the organ level could infer tumor evolution in space and time, affording more biological hypotheses.** Our study provides insights into ~~intrapatient~~ lesion-specific phenotypic heterogeneity in mCRC and yields substantial implications for designing metastasis-specific therapeutics.

3. Project Data Sphere is mentioned in beginning of results, but not explained in sufficient detail in the method section.

Revision: Methods Page 18, Line 371-373: All datasets are accessible in the Project Data Sphere, **an open-access platform that aggregates cancer clinical trial data from biopharmaceutical companies, academic medical centers, and government organizations** (<https://www.projectdatasphere.org/>).

4. In the results, it is not clearly stated which lesions were measured, how many (proportion of those found) were measured, which radiological method was used, etc.

Revision: Results Page 5, Line 85-87: The total target lesions were 19,180 with 94,174 radiographic measurements **by CT scan**, and there were 18,594 non-target lesions and 2,838 new lesions ~~with response status over time~~ **that had records of response status, appearance time, and anatomical site, and these lesions did not have longitudinal radiological measurements.**

5. line 75: the rationale behind using a cohort of head and neck squamous cell carcinoma makes little sense and needs to be clearly stated and motivated. Why use a cohort of squamous cell carcinoma to validate a cohort with adenocarcinoma, two very distinct different cancer types

Response: We agree that HNSCC is a very distinct cancer compared to CRC. The reason for us to choose HNSCC is to show that the anatomical distribution of lesion response and progression is beyond tumor biology, which is closely related to tissue microenvironments. We chose study NCT00460265 because this cohort of patients had similar metastases profiles and treatments with many patients in the training cohort (panitumumab plus chemotherapy). The anatomical distribution of lesion response and progression were quite consistent between the training and validation cohort. We made some edits to our text to explain more details about our validation dataset:

Revision: Methods Page 18, Line 370: Multiple mCRC ~~and mHNSCC~~ studies with longitudinal measurements...

Methods Page 18, Line 377-380: **Data from a Phase III study on panitumumab plus chemotherapy for mHNSCC were collected from Project Data Sphere and were analyzed with same workflow for validation. The clinical trials information was provided in Supplementary Table 5.**

6. line 81” recapitulated with a widely adopted growth model”: completely unclear what this means.

Revision: In the **Results Page 5, Line 94-95:** The tumor growth dynamics of 19,180 target lesions with 94,174 radiographical measurements were ~~recapitulated~~ **fitted with using a widely adopted tumor growth model²².**

7. similarly, ”the fraction of non-responding cells” – what kind of radiologically identified cell would that be? Maybe trivial for someone in the field, but not for most readers

Response: The non-responding cell population is the fraction of residual cell population during therapy. Most speculate this population is the resistant cells. However, the exact molecular features remain to be defined.

8. “the model adequately recapitulated the longitudinal profiles of tumour radiographic” – adequate in relation to what? Measured against what? Judged how?

Response: We compared tumor radiographic measurements (observations) with our model predictions at the individual lesion level. And the goodness of model fittings was provided in the **Supplementary Page 6, Line 38-47,** Supplementary Fig. 1. We applied several techniques to assess model predictive performance, including goodness of fitting plot, visual predictive check, quantification of residual error (vs. time and vs. observation). All analyses showed adequate model performance. In Supplementary Fig. 1a, the observations vs. predictions points were lying by $y=x$ line, which means the model well captured tumor radiographic measurements. In Supplementary Fig. 1c, we randomly picked 8 lesions, and their model predictions well captured observations.

9. line 90: lesions in the bone will give different results as they are much harder (at least very different from soft tissue lesions) to assess, particularly to call regression (rather than sclerosis).

Revision: We agree and acknowledged this in the **Discussion Page 17, Line 355-358: Our study has limitations. First, the size of metastases was measured by radiological CT scans, but lesions in the bone are generally hard to assess, which could result in quantification bias and variability.**

10. line 117, how were skin lesions assessed by radiology?

True “Skin” metastases are very rare as a manifestation of mCRC. Do the authors mean cutaneous metastases?

Response: The skin lesions were labeled as “Subcutaneous mass/nodule” and were measured by CT scan.

11. line 128. “Patients with these metastases, particularly those with low-high phenotype, had much worse survival outcomes (OS median 378 days vs. 561 days, $p<0.0001$)” – this is trivial (bad outcome for those with low response and high relapse), and it is unclear which group the 561 belong to?

Revision: In the **Results Page 8, Line 154-155:** Patients with these metastases, particularly those with low-high phenotype, had much worse survival outcomes **than those with other phenotypes...**

12. line 146: unclear that they switch back to mCRC

Response: In the **Results Page 9, Line 180: In mCRC, t**Treatment effects on organ-specific responses were also investigated.

13. line 115: “abdominal lesions”, what is included in that? Liver, spleen, peritoneum and abdominal lymph nodes are all abdominal organs...

Response: In the clinical trial, sponsors labeled the metastases in Peritoneal mass/nodule, Ascites, Gonadal mass/nodule as abdominal lesions and other lesions labeled as “Peritoneum”.

14. line 168: “Remarkably, despite comparable number of baseline metastases, patients whose first relapses were in the liver had a much worse prognosis than those whose first relapses were in lungs or other sites.”. “Number of metastases at baseline” is something that is never used in the clinic. What is interesting is what organ that is affected and how many different organs as that to some extent determines if the metastases are resectable.

It is also very well known that patients with “lung metastases only” do much better than metastases at other sites. That is why the presence of few and small lung metastases are not considered when assessing a patient for curative metastasectomy.

Response: We agree that the affected organs matter more than the number of metastases. To evaluate the anatomical site effect on patient prognosis, we compared patients with similar baseline metastases, even though the number of metastases do not impact prognosis as much as the affected organs. Unfortunately, only a small fraction of patients in these trials were undergone metastases resection and the metastatic resection did not seem to impact our analysis and results. Please find our comparison results in major comments #9.

15. line 248: “the other antibody panitumumab in our analyses did not significantly affect tumor regression rates either, suggesting its low direct cytotoxicity in patients” I believe the authors do not have the data to allow an analysis of such an effect of Panitumumab (as this would require an RCT).

Revision: We agree and deleted the speculative discussions. Please see major comment #8.

16. “Relapse” in brain metastases (in 67) – should that be “progression”? Or was there complete response (must be extremely rare) in brain mets for 67 patients? How does that fit to response in only 23? (Figure 3 a/b)

Revision: We changed “relapse” to “progression”. None of them had “complete response”, but “partial response” in 23 lesions.

17. Line 362: defining OS from start of therapies makes the results non-comparable with other studies. Either the start should be diagnosis of primary tumor or diagnosis of first metastatic lesion. If not, all sorts of bias are introduced. One could potentially compare someone who has undergone metastatic surgery and later treated with systemic therapy and then finally included in one of the mentioned trials to a patient diagnosed with mCRC and immediately included in one of the studies.

Same with PFS (defined as the start of therapies until RECIST-defined progression or death). The time with mCRC prior to inclusion in any of the studies must differ greatly.

Response: We agree that the surgical and treatment history could largely influence our analysis and results. Please see our response and additional analysis to tease out these confounding effect.

Revision: see major comments #2.

18. Table 1, line 520. The mean age indicate that the cohort consists of highly selected patients as normally, the mean age is around 70 year. Same with the gender distribution, less females than normally – why so? This needs to be addressed.

70% have undergone previous surgery. This is crucial for the interpretation of the entire study. Curative resection of primary only? Or metastatic resection? Must be clearly stated in order for the reader to interpret the results.

Response: We addressed this in major comments #5.

Figures

Figure 1b, line 523. What are abdominal metastases? Skin lesions more common than brain metastases? That does not correlate accurately to the clinical experience. Do you mean cutaneous metastases?

Response: We addressed this in minor comments #10 and #13.

Reviewer #2 (Remarks to the Author): Expert in colorectal cancer and computational genomics

Zhou et al. have modeled lesion-specific responses to systemic treatment across eight previously published clinical trials of metastatic colorectal cancer. Size measurements of multiple lesions and at multiple time points have been analyzed for each of 4,308 patients using a previously published method to model three different parameters, including the progression rate, regression rate and the fraction of non-responding cells. This is a comprehensive study revealing interesting features of organ-specific tumor growth dynamics during standard treatment of metastatic colorectal cancer, and should be of interest to the field. However, there are some issues that should be addressed.

Thank you for the positive comments.

1. The title, abstract and conclusion indicate that the study focuses on inpatient heterogeneity of treatment responses. With analysis of multiple lesions and multiple time points for each patient, data to analyze inpatient heterogeneity are available and this would be highly interesting. However, no direct measures of inpatient heterogeneity have been described or analyzed in the manuscript, and this does not seem to be a focus of the study. Instead, lesion-specific data are summarized per metastatic organ and compared across patients. I recommend rephrasing the text to more accurately reflect the purpose of the study.

Response: We agree with the reviewer. We modeled lesion-specific radiological data and performed statistical analyses at the organ level. We made the following edits in the manuscript:

Title Page 1, Line 1-2: Mapping ~~Inpatient~~ **Lesion-Specific** Response ~~Heterogeneity~~ and ~~Lesion-specific~~ **Relapse Progression Dynamics and Inter-Organ Variability** in Metastatic Colorectal Cancer

Abstract Page 2, Line 19-24: High ~~inpatient~~ **lesion-level response** heterogeneity persists, ... which are crucial to elucidating the mechanisms underlying ~~inpatient heterogeneity~~ **lesion-level variability**. However, our understanding of ~~inpatient~~ **lesion-specific response** heterogeneity on the macroscopic level, ... This study investigated ~~inpatient~~ **lesion-specific response** heterogeneity through ...

Abstract Page 2, Line 29-31: In conclusion, our study provides insights into ~~inpatient~~ **lesion-specific response and progression** heterogeneity in mCRC and creates impetus for metastasis-specific therapeutics.

Introduction Page 3, Line 38-39: ... but our understanding of such ~~inpatient~~ **lesion-level response** heterogeneity and its relevance to prognosis remains rudimentary.

Introduction Page 3, Line 40-46: Most investigations of ~~inpatient~~ **lesion metastatic** heterogeneity focus on ... underlying mechanisms of ~~inpatient~~ **lesion response** ... it is equivalently critical to study ~~inpatient~~ **metastatic** heterogeneity on the macroscopic level, ... a holistic view of **lesion-specific** ~~inpatient~~ heterogeneity across organs.

Introduction Page 4, Line 59-63: ~~Inpatient~~ **Lesion-specific** response heterogeneity is common ... have found that high ~~inpatient~~ response heterogeneity across organs is associated with ... Characterizing ~~inpatient~~ **lesion-specific** response heterogeneity in mCRC is valuable for prognosis and therapies.

Introduction Page 4, Line 75: Our study provides insights into ~~inpatient~~ **lesion-specific** phenotypic heterogeneity in mCRC...

Discussion Page 17, Line 363-367: In conclusion, we quantified ~~intrapatient~~ **lesion response and progression** heterogeneity by modeling the longitudinal size measurement of metastatic lesions ... to promote a more comprehensive view of ~~intrapatient~~ **lesion-specific** heterogeneity...

2. The clinical trials analyzed are a mix of studies of 1st and 2nd line treatment of metastatic colorectal cancer. The treatment line is a factor that needs to be accounted for in the analyses, considering its major impact on responses.

Response: We agree with the reviewer that the lines of treatment or the treatment history are potential confounding factors to our analysis.

To account for these confounding factors, we included treatment, together with patient age, BMI, gender, race, prior surgery (Yes or No), and line of therapy (first or second) as potential covariates in the random effect Cox proportional model for both progression and response dynamics. In the progression model, treatment, prior surgery history, and line of treatment are detected as significant covariates (p-val < 0.05), which are kept in the final model. In the response model, treatment, race, and line of treatment are significant so they are included in the final response model.

Despite statistically significant, the line of treatment was not found to greatly influence the rankings of hazard ratios across organs with a few exceptions. The treatment history switched the hazard ranking between the Pelvis and the Pancreas and between the Muscle/Soft tissues and the Adrenal. For response probability, the hazard ranking switched between the Pevis and Regional lymph nodes. The anatomical chart remains the same.

We also tested treatment, age, BSA, gender, race, prior surgery (Yes or No), prior radiation (Yes or No), and line of therapy (first or second) in the random effect Cox proportional model for both progression and response in mHNSCC patients (validation dataset). Treatment, age, prior radiation, and BSA were included in the final model for progression. Treatment and race were included in the response final model. The hazard rankings across organ remain unchanged for progression or response probability.

Revision: Results Page 7, Line 131-137 ~~Of note,~~ **We tested the covariate effects of treatment, age, BMI, gender, race, surgical history, and line of therapy (first or second line) in the random effect Cox proportional model. Significant covariates were included in the final model. Of note,** ~~treatment, surgical history, and line of therapy effects from either chemotherapy or combination therapy were included as a confounding factor~~ **covariates in the Cox regression progression model, and treatment, race, and line of therapy were included in the response model.**

Results Page 8-9, Line 170-174: **We built random effect Cox proportional models to estimate hazard ratios across organs in mHNSCC, as we did in mCRC. Treatment, age, radiation history, and BSA were included as covariates in the progression model. Treatment and race were considered as covariates for the response model.** The organ-specific hazard ratios for ~~relapse~~ **lesion progression** and response were ~~ranked shown in~~ **as we did in mCRC** (Supplementary Fig. 6c and 6d).

Methods Page 20, Line 420-424: **The potential covariate effects of treatment, line of therapy, age, gender, race, BMI, and surgical history were tested in mCRC patients and significant covariates (p-value < 0.05) were selected in the final model. For mHNSCC, we tested treatment, line of therapy, age, gender, race, BSA, and surgical/radiation history as potential covariates.**

Figure Page 35-37, Line 622-631, Figure 3 was updated:

We also noticed a typo in the manuscript that the bone progression probability should be the third lowest in mHNSCC. We updated that in Fig. 3 and Supplementary Fig. 6 (see above) and updated our conclusions about bone metastases in the Results Page 9, Line 174: In mHNSCC, metastases in the liver; bone; and brain also showed high relapse progression potential, in line with what we observed in mCRC.

3. The performance of the tumor growth dynamics models is evaluated by comparisons with observed tumor volume data, showing correspondence across lesions and in inter-lesion variances. These models are likely to provide a more dynamic response evaluation than the criteria used in the clinical trials, and it would be interesting to also compare directly the modeled and observed responses and relapses (the conclusions from the trials at defined time points). This could provide both a more intuitive evaluation of the models, and specify the potential advantages of the modeling.

Response: We agree with the reviewer that comparing the modeled and observed responses and progressions would be very helpful. We made two boxplots comparing model estimated response and progression rates with trial observations based on RECIST 1.0.

Revision: We included the results in the Supplementary Page 8, Line 56-62:

Supplementary Fig. 3. Model predicted progression or response duration by patient response status determined by RECIST 1.0. The progression (a) or response (b) times were predicted by tumor growth model and patient status of response were made per RECIST 1.0, including complete response (CR, n=118), partial response (PR, n=1,473), stable disease (SD, n=1,806), progressive disease (PD, n=781), and not evaluate (NE, n=130). The whiskers are drawn down to the 10th percentile and up to the 90th percentile. P-values were calculated by one-way ANOVA tests.

Results Page 7, Line 125-128: We compared our model-predicted lesion response and progression rates with patient response status per RECIST 1.1. Complete (CR) or partial responders (PR) had shorter time to response with more extended duration before progression than patients with stable disease (SD). (Supplementary Fig. 3)

4. The response parameters analyzed for prognostic associations should be evaluated in multivariable survival models including the parameters listed in Table 1, for characterization of prognostic value in relation to known prognostic factors.

Response: We thank the reviewer for the suggestion, which was addressed together with comment #2.

5. The number of metastatic organs should be considered in survival analyses of the four phenotypic features in supplementary fig 3. This would be particularly relevant to elucidate if the superior survival associated with metastases in high-low phenotype organs is because of the anatomical location of the metastases, or simply because the remaining patients have more extensive metastatic disease (involvement of more organs). It would also be interesting to see if results are similar in separate analyses of target lesions and non-target lesions.

Response: We tested lesion number as a covariate in the Cox proportional hazard model for patient overall survival together with organ phenotype (low-high or high-low). Lesion number is a statistically significant covariate but the organ phenotype is a stronger covariate, which remains significant after adjusting for lesion number. We made an adjusted survival plots for patients stratified by phenotype while matching lesion number (fix lesion number to 9 in both group, and 9 is the average lesion number in all patients):

We therefore concluded that organ phenotype is a significant predictor to patient survival.

Revision: We kept the original survival results in Supplementary Fig. 5.

6. The interpretation of the k=5 unsupervised clusters is somewhat confusing, in particular in survival analysis. There is presumably an overlap in characteristics among the groups, e.g. the “hetero-organ” group also has a substantial number of “liver-first” patients? To provide a clear analysis of the prognostic impact of the relapse sequence, it might be better to group the patients independent of the clustering analyses, that is, all patients with liver-first metastases versus the rest, etc. It is not surprising that patients with metastases to multiple organs have a poor survival, and this should rather be a stratification factor in prognostic analyses of the relapse sequence.

Response: We agree with the reviewer that it is a little confusing to name the cluster as “liver-first” “lung-first”, particularly when there were patients in “hetero-organ” group also having early relapse in the liver. The cluster names primarily reflect the most distinctive feature of the group, and other features in the group, such as the 2nd and 3rd progression organs, were also considered in the machine learning process. The machine learning was performed based on the whole progression orders rather merely on the first progression organ, and other features like affected organs and orders of organ progression were also important to the classification. Although it is not ideal, we named “Liver-first” or “Lung-first” to reveal the significant feature of that cluster. It is true that patients with metastases in multiple organs had the worst survival. Therefore, in the manuscript, we highlighted that our observations/conclusions were mostly made around the comparisons between patients with similar metastasis sites but different orders of progression across organs. In addition, we added more justifications for selecting optimal k in the revision. Please see details in response to Reviewer 4, comment #1.

7. Analyses are stratified according to treatment regimens, but only for chemo versus chemo plus antibodies. It is relevant to also compare chemo regimens FOLFOX versus FOLFIRI.

Response: This is a great point! In our random effect Cox model, we calculated the hazard ratios of all types of treatments.

Revisions: The results were summarized in the Supplementary Fig. 4 in the Supplementary Page 9, Line 63-65

Supplementary Fig. 4. Covariate effects on lesion response and progression. **a** and **b** rank the hazard ratios with 95% confidence interval of each potential covariate.

Results Page 7, Line 133-134: The covariate effects were summarized in Supplementary Fig. 4a and Supplementary Fig. 4b.

8. Results are summarized per metastatic organ and compared across patients, but is there variation among lesions per organ/patient? It would be interesting to evaluate the inpatient intra-organ heterogeneity in treatment response and relapse where possible.

Response: This is a great point! Our group has done analyses regarding the variation among lesions per organ and patients, and we found that response variations within organs are also informative to patient survival. Some of these results were published in our previous paper (Zhou J, Li Q, Cao Y. Cancer research, 2021, 81(9): 2522-2533.), which is cited in this manuscript References #18. This study is focused on lesion variations in response and progression at the organ level. In this study, we found phenotypic convergence at the organ level and the sequence of progression is very predictive of patient survival. These results have implications to lesion-specific treatment and shed light on tumor resistance and evolution in time and space.

Minor comments:

1. Some relevant details are missing from the description of the patient data, e.g. an overview of the number of lesions and time points evaluated per patient, as well as treatment doses and cycles and imaging modalities across studies. This info would indicate if stratified analyses by study are needed. Furthermore, the overview of clinical trials indicates that all studies were randomized phase III studies, but it appears that only one treatment arm was included for several. Please provide a description and justification of the patient selection.

Revision: We summarized lesion number and radiographic measurements per patient, and added this information into the Table 1. In addition, we added the line of therapy, patient surgical history, prior chemotherapy, and imaging modalities in the Supplementary Table 5. Unfortunately, we did not find individual patient dosing information in the study, so we did not consider the dose and pharmacokinetics factors in our analyses.

Revision: Related trial and patient information was added into **Tables Page 29-31, Line 601-603** Table 1 (see the updated Table below)

2. Table 1:

o It is stated that PFS and OS are reported in Table 1, but this information is missing. This would be helpful in the interpretation of the survival analyses.

Revision: We added median and IQR of the PFS and OS in Table 1.

o Patient numbers for treatment add up to more than 4308.

Revision: Corrected in the **Tables Page 29-31, Line 601-603:**

Updated Table 1 below. The corrected columns were highlighted in bold.

Variable	
Age, years (mean, sd)	60.2 (10.8)
Gender (n, %)	
Male	2538 (58.9)
Female	1770 (41.1)
Race (n, %)	
White/Caucasian	3883 (90.1)
Black/African American	104 (2.4)
Asian	142 (3.3)
Other	179 (4.2)
Body Mass Index, kg/m ² (mean, sd)	26.2 (5.1)
Tumor Type (n, %)	
Colon	2581 (59.9)
Rectal	1359 (31.5)
Unspecified	368 (8.5)
Prior Surgery (n, %)	
Yes	2993 (69.5)
No	1315 (30.5)

Prior Radiation (n, %)	
Yes	445 (10.3)
No	3345 (77.6)
Unknown	518 (12.1)
Treatment ¹ (n, %)	
Bevacizumab plus chemotherapy	376 (8.7)
Bevacizumab plus FOLFOX	630 (14.6)
FOLFIRI alone	1303 (30.2)
FOLFOX alone	762 (17.7)
Panitumumab plus Bevacizumab plus chemotherapy	372 (8.6)
Panitumumab plus FOLFOX	441 (10.2)
Panitumumab plus FOLFIRI	424 (9.8)
Response (n, %)	
Complete Response	118 (2.7)
Partial Response	1473 (34.2)
Progressive Disease	781 (18.1)
Stable Disease	1806 (41.9)
Not Evaluable	130 (3)
Metastatic organ number (n, %)	
1	553 (12.8)
2	1159 (26.9)
3	1146 (26.6)
≥4	1450 (33.7)
Metastatic lesion number per patient (median, IQR)	9 (6, 12)
Number of radiographic measurements per target lesion (median, IQR)	4 (3, 6)

KRAS status (n, %)	
Wild-Type	795 (18.4)
Mutant	593 (13.8)
Unknown	2920 (67.8)
Progression free survival, days (median, IQR)	247 (241.5)
Overall survival, days (median, IQR)	443 (410.3)

3. The survival analyses shown in supplementary figure 6 are not in line with the description in the text. Is the labeling correct in parts b and c?

Revision: In the **Results Page 10-11, Line 222-225:** Despite similar metastases, patients with first or second progression occurring in the liver had worse survival than those with early progression occurring in other organs (Supplementary Fig. 8b) **with no significant difference in the progression time (and Supplementary Fig. 8c).**

Supplementary Page 13, Line 87-97 Legend: Supplementary Fig. 8. ~~Liver relapse sequence~~ **The progression sequence of liver metastases within the Hetero-Organ group is associated with patient survival.**

Reviewer #3

The authors present a novel analysis with some intriguing findings further illustrating the heterogeneity of metastatic colorectal cancer. I have only a few comments, mostly regarding the presentation.

1. Some of the lesion sites included in the analysis are modeled based on very few observations (e.g., skin, pancreas, bone), leading to questions as to the robustness of the model for these sites. The authors should consider focusing their analysis on the 8-10 most represented sites.

Response: We thank the reviewer for this suggestion. We agree that the number of lesions in the skin, pancreas, and brain/CNS lesion numbers are relatively small compared to that in the liver, lung, and lymph nodes. In order to evaluate the robustness of our methods and results, we did another analysis by merging the organs with lesion number < 100 (pancreas, GR, kidney, muscle/soft tissue, brain/CNS, skin) into a group called “Small number” (n=307) and re-analyzed the organ-level tumor response and regression hazards.

In the figure above, we ranked the hazard ratios with 95% confidence interval (95% CI) in lesion response and progression across organs. GR, Genitourinary and Reproductive; CNS, central nervous system; GI, Gastrointestinal tract; LN, lymph nodes. By comparing this Figure with Fig. 3a and Fig. 3b (Figures Page 36, Line 620), it seems that the organs with small numbers of lesions would not noticeably impact hazard estimates and the ranking. We therefore kept these organs in our model for a broad assessment of anatomical sites. As shown in Figure 3a and 3b, organs with small numbers of lesions usually have wider 95% CI and relatively harder to reach statistical significance. Although small lesion numbers in some organs, they appear to be sufficient to reach statistical significance.

2. The use of a validation set that includes patients with metastatic cancers of a different primary site should be better justified.

Response: We agree that HNSCC is a very distinct cancer compared to CRC. The reason for us to choose HNSCC is to show that the anatomical distribution of lesion response and progression is beyond tumor biology, which is closely related to tissue microenvironments. We chose study NCT00460265 because this cohort of patients had similar metastases profiles and treatments with many patients in the training

cohort (panitumumab plus chemotherapy). The anatomical distribution of lesion response and progression were quite consistent between the training and validation cohort. We made some edits to our text to explain more details about our validation dataset:

Revision: Methods Page 18, Line 370: Multiple mCRC and mHNSCC studies with longitudinal measurements...

Methods Page 18, Line 377-380: Data from a Phase III study on panitumumab plus chemotherapy for mHNSCC were collected from Project Data Sphere and were analyzed with same workflow for validation. The clinical trials information was provided in Supplementary Table 5.

3. The objectively measured outcomes should be more heavily weighted in the authors' presentation than the model-estimated parameters.

Response: We agree with the reviewer on this point.

Revision: We compared our model-derived estimates with the clinical observations regarding the response status for each patient. We included the results in the Supplementary Page 8, Line 56-62:

Supplementary Fig. 3. Model predicted progression or response duration by patient response status determined by RECIST 1.0. The progression (a) or response (b) times were predicted by tumor growth model and patient status of response were made per RECIST 1.0, including complete response (CR, n=118), partial response (PR, n=1,473), stable disease (SD, n=1,806), progressive disease (PD, n=781), and not evaluate (NE, n=130). The whiskers are drawn down to the 10th percentile and up to the 90th percentile. P-values were calculated by one-way ANOVA tests.

Results Page 7, Line 125-128: We compared our model-predicted lesion response and progression rates with patient response status per RECIST 1.0 in the trials. Complete (CR) or partial responders (PR) had shorter time to response with more extended duration before progression than patients with stable disease (SD). (Supplementary Fig. 3)

Reviewer #4

The reviewer's background is in machine learning and bioinformatics and consequently the comments below apply to these aspects of the paper, rather than the more general discussion of inpatient response heterogeneity.

The paper was interesting to read, and well written. In terms of the data analysis and machine learning there were some shortfalls with the techniques used which need clarifying, and particularly, slightly sloppy descriptions of models: those familiar with the area would understand the approach used, though.

Thank you for the positive comments.

1. 158: 'Relapse sequence across organs predicts patient survival'. At this point the authors discuss the use of the Elbow method and the Silhouette method to find K, the number of proposed clusters in a K means clustering algorithm. The paper refers to Supplementary Figures 5A and 5B. Firstly, the caption below Figure 5 looks inadequate, simply saying the Elbow method and Silhouette method were used. I think it would help to provide some more detail here. Specifically, the Elbow method is a heuristic based on identifying the inflexion point of the curve. By sight this does look to be about 5. The Silhouette method is based on the similarity of a datapoint to its own identified cluster and dissimilarity to other proposed clusters. In this context it is not particularly clear from Figure 5B that K=5 is a suitable choice, at least better than other choices? Similar remarks apply in the area of l. 177 where Figure 5C and 5D are discussed.

Determining the optimal number of clusters in the data is something of a 'can of worms' issue since it involves avoiding fits to noise, the dual objectives of maximising intra-cluster similarities and inter-cluster dissimilarities, and possible variant methods, e.g. mixed membership models (data points can potentially overlap several clusters), may give a better model.

Response: This is all great comments and we addressed this question together with the next one.

2. In short, I think the approach should be: (a) a variety of methods are used to determine K and an objective discussion is given of the concordance or discordance between these different methods, (b) you may have to run with a model which is not unambiguously justified (in terms of the best choice of K). In that case the model has to be explanatory and, in particular, justified on independent held-out data which validates the choice of K. For (a) I think it could be interesting to have a look at other clustering criteria, such as the Akaike information criterion (AIC) or Bayesian information criterion (BIC): do they also indicate K=5? If not, it is appropriate to objectively discuss the concordance and discordance between criteria and go with a model which would appear to work best according to be (b). Certainly, the K(=5) (l.161) isn't properly justified.

Response: We thank the reviewer for pointing this out. We selected the optimal K based on the Elbow and Silhouette methods, AIC/BIC, as well as the clinical implications of each cluster. We aimed to use this classification to predict patient survival, and the classification was selected to best separate patient survival profiles with distinct and interpretable features. In Supplementary Fig 5a and 5b, it seems that the data could be better separated with more clusters. And models with 4 or more clusters showed modest improvement in BIC and AIC. We also compared the concordance of the survival curves separated by 3, 4, 5, or 6 clusters. K=5 has the highest concordance ($c=0.5901$) compared to 3 ($c=0.5709$), 4 ($c=0.5833$), 5 ($c=0.5901$), and 6 ($c=0.585$). Therefore, we chose five clusters to separate patients with interpretable

features of their progression patterns. Same selection criteria were also applied to Hetero-Organ patients. $k=4$ has higher concordance ($c=0.5556$) than $k=3$ ($c=0.5435$) in survival curves separation.

We thank the reviewer for the suggestion of using AIC and BIC criteria to evaluate different k . We calculated AIC and BIC across different k in both all-patients and Hetero-Organ patients. We added these results to Supplementary Fig. 7e and Supplementary Fig. 7f, Page 12, Line 79-85:

Revision:

Figure legend: e and f Akaike information criterion (AIC) and Bayesian information criterion (BIC) across different k for all the patients (e) or Hetero-Organ patients (f).

We added the following texts to the Results Page 9-10, Line 195-201: Elbow sum of square³³ (Supplementary Fig. 7a) and Silhouette score³⁴ (Supplementary Fig. 7b) were calculated to determine the optimal k ($=5$) in the final classification. **Akaike information criterion (AIC) and Bayesian information criterion (BIC)³⁵ were also applied to find optimal k (Supplementary Fig. 7e). $K=4, 5,$ and 6 showed similar performance in the model evaluation metrics. We also compared the patient survival profiles using concordance. The select $k=5$ has the finest separation of patient survival, resulting in distinct features of progression for each group. Five groups of patients were thus identified...**

Here citation 35 were added to explain AIC and BIC in the References Page 25, Line 528-529: Burnham, K. P. & Anderson, D. R. Multimodel inference: understanding AIC and BIC in model selection. *Sociol. Methods Res.* 33, 261–304 (2004).

Results Page 10, Line 219-220: Four groups of patients were optimally clustered (Supplementary Fig. 7c, and 7d, and 7f),

Supplementary Fig. 7e and Fig. 7f indicated that the model performance is similar between $k=4, 5,$ or 6 . We decided to keep $k=5$ in our results since it can best separate patient survival profiles. In order to better explain the Elbow and Silhouette methods and AIC and BIC metrics and justify $k=5$, we added the following texts in the Methods Page 20, Line 430-435: Elbow method³³ and Silhouette score³⁴, as well as AIC and BIC³⁵, were applied to find optimal k . **The Elbow method selects optimal k based on the inflexion point of the performance curve. The Silhouette method is based on the similarity of a data-point to its own identified cluster and dissimilarity to other clusters. The AIC and BIC provide complementary measures that balance model complexity and predictive performance. The optimal k was also selected to yield adequate separation of patient survival and interpretable features of lesion progression. The relapse patterns of patients clustered with different k were also compared to help determine the choice of k in the final classification.**

3. 202. 'Machine learning model predicts lesion relapse sequence'. Though I can't comment on the meaningfulness of the identified categories, gradient boosting is a good choice here and the ROC curves in Figure 8 indicate a reasonable predictive model.

Response: We thank the reviewer for the comment.

4. 319. 'for stochastic approximation expectation maximisation': a citation is suitable here (there are various models of this type).

Response: We added a citation to the **Methods Page 19, Line 394:** Stochastic approximation expectation-maximization (SAEM) algorithm⁵⁷ was applied to search global optimum...

References Page 27, Line 577-578: 57. Savic, R. & Lavielle, M. Performance in population models for count data, part II: a new SAEM algorithm. *J. Pharmacokinet. Pharmacodyn.* 36, 367–379 (2009).

5. 343. 'Relapse pattern classification and prediction'. Some of the description here is a little unclear, though correct. First sentence might be better phrased 'We used the K means clustering algorithm, from machine learning, to categorise all the patients based on their organ relapse.' ['categorise' sounds better than 'classify' since the latter is a bit associated with supervised learning in the context of machine learning]. In the next sentence they should provide citations after Elbow method and Silhouette score. In this paragraph an issue which was not covered was if data input attributes coming into these models were standardised or normalised in some way. If attributes are on very different scales this would affect the performance of a clustering or machine learning algorithm. I'm not sure if the authors applied their own standardisation or, quite possibly, data standardisation was incorporated within the algorithms and packages which they used? However, this is an issue which should be at least briefly mentioned.

Response: We thank reviewer for the suggestions. We made the following edits.

Revision:

Page 20, Line 427: We used the k-means ~~machine learning~~ **clustering** algorithm to ~~classify~~ **categorize** all the patients...

We added how we standardized variables in Gradient Boosting model in **Page 21, Line 445-447:** Continuous predictors were normalized and categorical predictors were transformed to dummy variables **using "OneHotEncoder" package "softmax" function.**

Page 21, Line 439-440: The research samples were randomly split into a-training and testing ~~groups~~ **sets** at a ratio of 4:1, **with rotation of the test dataset to implement 5-fold cross validation.**

Page 21, Line 438-439: The model outcome is the patient ~~relapse~~ **progression** sequence classified in k-means **clustering** algorithm. Model ~~predictors~~ **inputs** included patient clinical and demographic characteristics...

We added citation 32 to explain the unsupervised k-means clustering model in the **Results Page 9, Line 194:** We built a k-means unsupervised clustering model³²... and the **Methods Page 20, Line 428:** We used the k-means ~~machine learning~~ **clustering** algorithm³²...Please find citation 32 in **References Page 24, Line 520-521:** 32. Sinaga, K. P. & Yang, M.-S. Unsupervised K-means clustering algorithm. *IEEE access* 8, 80716–80727 (2020).

We added citation 36 to explain gradient boosting model in the **Method Page 21, Line 437:** Gradient Boosting algorithm³⁶ was applied to build a relapse pattern predictive model...Please find citation 36 in

the **References Page 25, Line 530**: 36. Natekin, A. & Knoll, A. Gradient boosting machines, a tutorial. *Front. Neurobot.* 7, 21 (2013).

6. 350. 'into a training and testing groups at a ratio of 4:1'. In the follow on sentence is it is apparent that the authors have simply used five-fold cross validation, however, this sentence does not exclude the possibility that the algorithm was trained on 80% of the data and a single held out 20% of the data. Perhaps a better wording might be 'the research samples were randomly divided into training test set according to the ratio of 4:1, with rotation of the test set data to implement 5-fold cross validation.'

Response: We thank reviewer for the suggestions. **Methods Page 21, Line 438-439**: The research samples were randomly split into a-training and testing ~~groups-sets~~ at a ratio of 4:1, **with rotation of the test dataset to implement 5-fold cross validation.**

7. 353. K means algorithm-> K means clustering algorithm? The next sentence begins 'Model predictors included patient clinical and demographic characteristics' : it is not clear to the reader from this phrasing if you are referring to the inputs to the model (the attributes or features used) or the outputs from the model, this should be clearly stated.

Revision: We made the following edits to the manuscript:

In the **Methods Page 21, Line 441-442**: The model outcome is the patient relapse sequence classified in k-means **clustering** algorithm. Model ~~predictors~~ **inputs** included patient clinical and demographic characteristics...

Other revisions:

Since we added two figures in the Supplementary Materials (Supplementary Fig. 3 and Supplementary Fig. 4), the labels of other supplementary figures were changed:

In **Results Page 5, Line 92**: The data was also from Project Data Sphere with similar criteria as CRC data (Supplementary Fig. ~~3a6a~~).

In **Results Page 8, Line 157**: ... (OS median 378 days vs. 561 days, $p < 0.0001$, Supplementary Fig. ~~3a5a~~).

In **Results Page 8, Line 161**: ... (OS median 770 days vs. 524 days, $p < 0.0001$, Supplementary Fig. ~~3b5b~~).

In **Results Page 8, Line 169**: ... eleven metastatic organs (Supplementary Fig. ~~4a-6a~~ and ~~4b6b~~).

In **Results Page 8-9, Line 173-174**: ... ~~ranked shown in~~, as we did in mCRC (Supplementary Fig. ~~4e 6c~~ and ~~4d6d~~).

In **Results Page 9, Line 196**: ... Elbow sum of square³³ (Supplementary Fig. ~~5a7a~~) and Silhouette score³⁴ (Supplementary Fig. ~~5b-7b~~)...

In **Results Page 10, Line 220**: ... patients were optimally clustered (Supplementary Fig. ~~5e7c~~, and ~~5d7d~~, and ~~7f~~),

In **Results Page 10-11, Line 222-224**: ... order of liver lesions (Supplementary Fig. ~~6a8a~~). ... occurring in other organs (Supplementary Fig. ~~6b8b~~) with no significant difference ... (and Supplementary Fig. ~~6e8c~~).

In **Results Page 11, Line 239-243**: ... overall survival (Supplementary Fig. ~~7a9a~~) compared to chemotherapy alone, ... two treatment groups (Supplementary Fig. ~~7b9b~~). ... especially for those who had early liver lesion ~~relapse~~ **progression** (Supplementary Fig. ~~7e-9c-h~~).

In **Results Page 12, Line 251-255**: ... which indicated fair performance (Supplementary Fig. ~~8a10a~~). ... of the latter three groups (Supplementary Fig. ~~8b10b~~).

REVIEWER COMMENTS

Reviewer #1 (Remarks to the Author):

Thank you for the opportunity to read your revised version. I started out by reading the full manuscript without tracked changes and perceived it as much more coherent and comprehensible – even to a clinician – for which I congratulate you.

Could you please clarify if having the primary tumor not resected (as about 2600 patients ought to have not based on supple table 5) affected progression dynamics? Clinically a very important and interesting question!

Maybe you should also add to the limitation section that the lesion-Specific response and progression dynamics that you have reported on applies to a mostly palliative cohort.

In addition, I only have some minor comments

Line 231 (and 387) BSA, abbreviation without an explanation when used for the first time in text

Supplementary Fig. 6a is cited as the first figure, maybe it then should be re-named to suppl fig 1a?

Check the wording of the long sentence starting at line 269

Table 1: what is “Prior Surgery (n, %”, surgery of primary tumor? Metastatic surgery? Please specify and sub-divide into "surgery for primary" and "surgery for metastases"

Table 1: Prior Radiation (n, %)? Radiation is only indicated if rectal cancer. Do you mean 445 out of the 1359 with rectal cancer? or was it radiation of lung metastases?

In table 1, is OS and PFS really median with IQR or have you reported means with sd as you have introduced Sd in the text below the figure. My guess is that median survival is more appropriate as it normally is non-normally distributed.

Figure 5, what is TAR

Reviewer #2 (Remarks to the Author):

I thank the authors for the revision, which has addressed most of my comments. I think the manuscript has improved. There is one outstanding issue related to my previous comment 4 (multivariable survival analyses).

I appreciate that covariables have been included in the Cox models for prediction of lesion response or progression (in response to previous comment 2). However, this does not address the concern that several of the parameters listed in Table 1 are potential confounding factors also in subsequent survival analyses. A similar testing of the effect of covariables (and inclusion in a multivariable survival model if needed) is appropriate also in relation to the prognostic value of organ-specific response and progression probability patterns (illustrated in Supplementary Figure 5) and the organ-level progression sequence (Figure 4b). This would strengthen the prognostic data of the study.

I also recommend changing the phrase “response/progression time” to the more appropriate “time to response/progression”.

Reviewer #3 (Remarks to the Author):

No further comments

Reviewer #4 (Remarks to the Author):

The authors have improved the paper as regards the machine learning aspects, with improved wording and a more consistent approach. The authors implement AIC and BIC and new supporting supplementary figures 7e and 7f. The results still look problematic in that 'The selected $k=5$ has the finest separation of patient survival, resulting in distinct features of progression for each group': if searching across different k this will inevitably occur (potential overfitting, no real supported effect). I am not sure if it is possible to support inferences with any further new/alternate datasets?

Manuscript NCOMMS-22-09853

Title: Mapping Lesion-Specific Response and Progression Dynamics and Inter-Organ Variability in Metastatic Colorectal Cancer

RESPONSES TO CRITIQUES

We would like to thank both reviewers as well as the editor for their careful reading of our paper and their thoughtful suggestions and comments. We have now addressed all comments in a revised version of the manuscript. Please find below a detailed point-by-point response to all issues raised, in which all major revisions are highlighted by **bold font** and deleted text shown with strikethroughs (~~deleted text~~) in the copy of the submission (manuscript with marked revisions). **The Page and Line # are highlighted in red.**

REVIEWER COMMENTS

Reviewer #1 (Remarks to the Author):

Thank you for the opportunity to read your revised version. I started out by reading the full manuscript without tracked changes and perceived it as much more coherent and comprehensible – even to a clinician – for which I congratulate you.

Thank you!

Could you please clarify if having the primary tumor not resected (as about 2600 patients ought to have not based on supple table 5) affected progression dynamics? Clinically a very important and interesting question!

Maybe you should also add to the limitation section that the lesion-Specific response and progression dynamics that you have reported on applies to a mostly palliative cohort.

Response: We thank the reviewer for the comments. We agree with the reviewer that primary surgical resection makes large difference in patient prognosis and tumor progression. Indeed, when we compared lesion time to progression, surgical history was a significant covariate. In the **Results session Page 7, Line 126-128:** Of note, treatment, **surgical history**, and line of therapy were included as significant covariates in the progression model...

We also added patient prior surgical history detailed information in **Page 28, Table 1** (as minor comments pointed out):

Prior Surgery (n, %)	
Yes	2993 (69.5)
Primary tumor resected	2663 (61.2)
Liver metastases resected	261 (6.05)¹
Lung metastases resected	27 (0.6)¹
Lymph nodes metastases resected	206 (4.8)¹
No	1315 (30.5)

In addition, I only have some minor comments

Line 231 (and 387) BSA, abbreviation without an explanation when used for the first time in text

In the **Results session Page 8, Line 158-159**: ... and **body surface area (BSA)** ...

Supplementary Fig. 6a is cited as the first figure, maybe it then should be re-named to suppl fig 1a?

Response: We removed this cite for clarity.

Check the wording of the long sentence starting at line 269

Response: We reworded the sentence for better readiness:

In **Page 14, Line 273**: ... was more closely determined by the progression rates rather than the response rate-. **This finding was** in line with Stein et al.,

Table 1: what is "Prior Surgery (n, %)", surgery of primary tumor? Metastatic surgery? Please specify and sub-divide into "surgery for primary" and "surgery for metastases"

Response: We specified the surgical history in **Page 28, Line 570**:

Prior Surgery (n, %)	
Yes	2993 (69.5)
Primary tumor resected	2663 (61.2)
Liver metastases resected	261 (6.05)¹
Lung metastases resected	27 (0.6)¹
Lymph nodes metastases resected	206 (4.8)¹
No	1315 (30.5)

¹Study NCT00272051 and NCT00305188 did not specified the site of the resected metastases and were thus not included in the Table.

Table 1: Prior Radiation (n, %)? Radiation is only indicated if rectal cancer. Do you mean 445 out of the 1359 with rectal cancer? or was it radiation of lung metastases?

Response: In our dataset, 49 patients with colon cancer had prior radiation history for lung metastases. The details of radiation history were not well documented in the clinical trial data. The other patients were either with rectal cancer or without clear distinction of colon and rectal cancer.

Revision: we specified radiation in **Page 30, Line 573**: ²**Radiation was applied to rectal cancer and some colon cancer with lung metastases.**

In table 1, is OS and PFS really median with IQR or have you reported means with sd as you have introduced Sd in the text below the figure. My guess is that median survival is more appropriate as it normally is non-normally distributed.

Response: We reported by median with IQR. We labeled them as Progression free survival, days (median, IQR) and Overall survival, days (median, IQR).

Figure 5, what is TAR

Revision: In the **Page 37, Line 619:** TAR + Chemo, antibody targeted therapies (**bevacizumab or panitumumab**) plus chemotherapy;

Reviewer #2 (Remarks to the Author):

I thank the authors for the revision, which has addressed most of my comments. I think the manuscript has improved. There is one outstanding issue related to my previous comment 4 (multivariable survival analyses).

Thank you.

I appreciate that covariables have been included in the Cox models for prediction of lesion response or progression (in response to previous comment 2). However, this does not address the concern that several of the parameters listed in Table 1 are potential confounding factors also in subsequent survival analyses. A similar testing of the effect of covariables (and inclusion in a multivariable survival model if needed) is appropriate also in relation to the prognostic value of organ-specific response and progression probability patterns (illustrated in Supplementary Figure 5) and the organ-level progression sequence (Figure 4b). This would strengthen the prognostic data of the study.

Response: We thank the reviewer for pointing this out. To assess the confounding factors shown in Figure 4b, we built a Cox proportional hazard model predicting patient overall survival, and incorporated patient progression clusters and many other potential confounders listed in Table 1 into the model. We evaluated the following confounders to patient survival: age, gender, race, treatment type, tumor type, prior surgery history, prior radiation history, BMI, KRAS type, metastatic organ number, and metastatic lesion number. In the presence of these confounders, patient progression pattern remains as significant covariates to patient survival:

The hazard ratio of Liver-First on overall survival is significantly higher than Lung-First and Other-First ($p < 0.001$). And the hazard ratio of Mono-Organ is significantly lower than Hetero-Organ. The results were consistent with Figure 4b.

We performed similar analyses as shown in Supplementary Figure 5. We built Cox proportional models to patient overall survival and incorporated patient Low-High phenotype/High-Low phenotype and age, gender, race, treatment type, tumor type, prior surgery history, prior radiation history, BMI, KRAS type, metastatic organ number and metastatic lesion number as confounding factors. Patients with low-high phenotype metastases (genitourinary and reproductive system, adrenal, muscle/soft tissues, bone, brain/central nervous system) still have significantly worse survival than patients without low-high phenotype metastases (**HR = 1.21, $p < 0.001$**). Patients only have metastases in high-low phenotype organs

(lymph nodes, chest, spleen, lung) still have significantly better survival than those with other phenotypes (**HR = 0.76, p<0.05**).

I also recommend changing the phrase “response/progression time” to the more appropriate “time to response/progression”.

In **Page 6, Line 114**: ~~Response and progression times~~ **Time to response and progression** were then derived...

In **Page 7, Line 120-121**: We integrated the ~~response time~~ **time to response** for both target and non-target lesions and the ~~progression time~~ **time to progression** for all lesions, ...

In **Page 10, Line 210**: but two groups showed no significant difference in subsequent ~~progression times~~ **time to progression**...

In **Page 11, Line 223**: The average ~~progression times~~ **time to progression** were much longer...

In **Page 11, Line 228**: Patients’ progression patterns and lesion ~~progression time~~ **time to progression** ...

In **Page 18, Line 377**: Tumor ~~response and progression times~~ **time to response and progression**

In **Page 18, Line 380**: Target lesion ~~response time~~ **time to response** (when the lesion volume decreases \geq 20% from baseline)...

In **Page 18, Line 381**: ... and ~~progression time~~ **time to progression** (when the lesion volume increases \geq 30% from tumor nadir or at least 200 mm³ increase from nadir)...

In **Page 19, Line 385-386**: The ~~progression time~~ **time to progression** for new lesions were defined as the detection time.

Reviewer #4 (Remarks to the Author):

The authors have improved the paper as regards the machine learning aspects, with improved wording and a more consistent approach. The authors implement AIC and BIC and new supporting supplementary figures 7e and 7f.

Thank you.

The results still look problematic in that 'The selected k=5 has the finest separation of patient survival, resulting in distinct features of progression for each group': if searching across different k this will inevitably occur (potential overfitting, no real supported effect). I am not sure if it is possible to support inferences with any further new/alternate datasets?

Response: We agree with the reviewer that it is possible that searching across different k could cause overfitting. To resolve this potential issue, we found biological evidence to support our finding. For instance, in one of our previous papers (Zhou J, Li Q, Cao Y. Spatiotemporal heterogeneity across metastases and organ-specific response informs drug efficacy and patient survival in colorectal cancer. *Cancer research*, 2021, 81(9): 2522-2533), we found the favorable responses in liver metastases for colorectal cancer patients (> 30% shrinkage) were associated with extended patient survival ($P < 0.001$), in contrast to lesions in the lungs and lymph nodes. We also agree that new/alternative datasets would be valuable to support our inference, we included this in our model limitation.

Revision: In **Discussion, Page 16, Line 330-332: The machine learning identified distinct patterns of tumor progression but it has a potential overfitting issue. More validations of our findings using additional datasets are warranted.**

REVIEWERS' COMMENTS

Reviewer #1 (Remarks to the Author):

I once again thank the authors for their quick revision and the editor for again letting me read the manuscript.

My comments and questions are answered and I perceive that the authors have made significant changes which have improved the manuscript.

Reviewer #2 (Remarks to the Author):

I thank the authors for sharing data from multivariable survival analyses supporting an independent prognostic value of the variables highlighted in Supplementary Figure 5 and Figure 4. This strengthens the prognostic analyses and I recommend including these new data in the manuscript.

I have no further comments.

Reviewer #4 (Remarks to the Author):

From reviewer #4:

Suggested rewording of:

The machine learning identified distinct patterns of tumor progression but it has a potential overfitting issue. More validations of our findings using additional datasets are warranted.

to(?):

Machine learning has therefore identified distinct patterns for tumor progression, but the future validation of these findings will be through acquisition and study of further independent datasets.

Manuscript NCOMMS-22-09853

Title: Mapping Lesion-Specific Response and Progression Dynamics and Inter-Organ Variability in Metastatic Colorectal Cancer

RESPONSES TO CRITIQUES

We would like to thank both reviewers as well as the editor for their careful reading of our paper and their thoughtful suggestions and comments. We have now addressed all comments in a revised version of the manuscript. Please find below a point-by-point response to all issues raised, the text revised in the manuscript were highlighted.

Reviewer #2 (Remarks to the Author):

I thank the authors for sharing data from multivariable survival analyses supporting an independent prognostic value of the variables highlighted in Supplementary Figure 5 and Figure 4. This strengthens the prognostic analyses and I recommend including these new data in the manuscript.

I have no further comments.

Response: We added the hazards ratios of different patient cluster in overall survival Cox proportional models adjusting confounding factors in Supplementary figure 8 (Supplementary Information Page 13). We also added our analyses results to the legends of Supplementary Figure 5 (Supplementary Information Page 10). We thank the reviewer again for the great comments and efforts.

Reviewer #4 (Remarks to the Author):

From reviewer #4:

Suggested rewording of:

The machine learning identified distinct patterns of tumor progression but it has a potential overfitting issue. More validations of our findings using additional datasets are warranted.

to(?):

Machine learning has therefore identified distinct patterns for tumor progression, but the future validation of these findings will be through acquisition and study of further independent datasets.

Response: Corrected in Page 16, Line 328-330.